# Mechanistic Advances in Hypoglycemic Effects of Natural Polysaccharides: Multi-Target Regulation of Glycometabolism and Gut Microbiota Crosstalk

**DOI:** 10.3390/molecules30091980

**Published:** 2025-04-29

**Authors:** Liquan Zhou, Jiani Li, Chen Ding, Yimiao Zhou, Zuowei Xiao

**Affiliations:** 1Homologous Innovation Laboratory of Medicine and Food, Hunan University of Chinese Medicine, Changsha 410208, China; zlq8067@126.com (L.Z.); 20243807@stu.hnucm.edu.cn (J.L.); 20243738@stu.hnucm.edu.cn (C.D.); 2Hunan Engineering and Technology Research Center for Health Products and Life Science, Hunan University of Chinese Medicine, Changsha 410208, China; 3School of Pharmacy, Hunan University of Chinese Medicine, Changsha 410208, China; 4Xiangyin Campus, Xiangxing College, Hunan University of Chinese Medicine, Yueyang 414615, China

**Keywords:** natural polysaccharides, hypoglycemia, mechanism of action, gut microbiota, signaling pathway

## Abstract

Natural polysaccharides (NPs), as a class of bioactive macromolecules with multitarget synergistic regulatory potential, exhibit significant advantages in diabetes intervention. This review systematically summarizes the core hypoglycemic mechanisms of NPs, covering structure–activity relationships, integration of the gut microbiota–metabolism–immunity axis, and regulation of key signaling pathways. Studies demonstrate that the molecular weight, branch complexity, and chemical modifications of NPs mediate their hypoglycemic activity by influencing bioavailability and target specificity. NPs improve glucose metabolism through multiple pathways: activating insulin signaling, improving insulin resistance (IR), enhancing glycogen synthesis, inhibiting gluconeogenesis, and regulating gut microbiota homeostasis. Additionally, NPs protect pancreatic β-cell function via the nuclear factor E2-related factor 2 (Nrf2)/Antioxidant Response Element (ARE) antioxidant pathway and Toll-like receptor 4 (TLR4)/nuclear factor-κB (NF-κB) anti-inflammatory pathway. Clinical application of NPs still requires overcoming challenges such as resolving complex structure–activity relationships and dynamically integrating cross-organ signaling. Future research should focus on integrating multi-omics technologies (e.g., metagenomics, metabolomics) and organoid models to decipher the cross-organ synergistic action networks of NPs, and promote their translation from basic research to clinical applications.

## 1. Introduction

Diabetes mellitus, a global chronic metabolic disease, has nearly quadrupled in prevalence over the past three decades [1]. The disease is characterized by defective insulin secretion (INS) or IR, and long-term hyperglycemia can lead to serious complications such as cardiovascular disease, nephropathy, retinopathy, etc., which greatly threaten human health. Although the commonly used drugs (e.g., sulfonylureas and biguanides) can control blood glucose in the short term, long-term use of these drugs may lead to hypoglycemia, weight gain, liver and kidney toxicity, and drug resistance. Therefore, the development of safe and efficient new intervention strategies has become an important topic in diabetes research [2].

NPs are naturally synthesized macromolecular carbohydrates consisting of more than 10 monosaccharides linked by glycosidic bonds, with a molecular weight of more than 10 kDa, which are widely found in plants, animals, fungi, and microorganisms, such as cellulose, β-glucan, alginate, etc. NPs have functions such as structural support, energy storage and regulation of biological activity. For example, cellulose can provide mechanical support for plants, plant starch and animal glycogen can provide energy, and mulberry leaf polysaccharide has hypoglycemic activity.

Recent studies have shown that NPs not only have the advantages of wide availability, high biocompatibility and low toxicity, but also can synergistically regulate glucose metabolism through multiple pathways such as improving INS sensitivity, protecting pancreatic β-cells and inhibiting chronic inflammation, as a result of which they are regarded as a breakthrough candidate for diabetes intervention [3]. For example, Chen et al. [4] investigated the ameliorative effect of *Enteromorpha prolifera* polysaccharide on glycemic abnormalities in Zucker diabetic obese rats. The results showed that *Enteromorpha prolifera* polysaccharide significantly reduced fasting blood glucose (FBG), fasting INS, glycated hemoglobin and IR index, and also enhanced INS sensitivity and beta cell function. Yang et al. [5] investigated the hypoglycemic effect and mechanism of *Hovenia dulcis* (Guaizao) polysaccharide on high-fat and high-sugar diet combined with streptozotocin (STZ)-induced type 2 diabetes (T2D) rats. Oral administration of Guaizao polysaccharide significantly increased body weight and hepatic glycogen, lowered FBG, alleviated hyperinsulinemia and IR, and ameliorated lipid metabolism disorders. Guaizao polysaccharide was able to repair hepatic injury by alleviating hepatic oxidative stress, and at the same time elevated the levels of short-chain fatty acids (SCFAs). Guaizao polysaccharide was able to repair hepatic damage by activating phosphorylated fatty acids, and at the same time elevated levels of SCFAs. Guaizao polysaccharide improves glucose metabolism by activating the phosphatidylinositol 3-kinase (PI3K)/protein kinase B (Akt) signaling pathway, up-regulating the expression of insulin receptor (INSR), PI3K, Akt, and glucose transporter 4 (GLUT4), and regulating glycogen synthase (GS), glycogen synthase kinase 3β (GSK-3β), and Forkhead box protein O1 (FoxO1). Guaizao polysaccharide also regulates glucose-6-phosphatase (G6Pase) and phosphoenolpyruvate carboxykinase (PEPCK) through the AMP-activated protein kinase (AMPK) pathway.

Unlike monosaccharides or oligosaccharides, most NPs are difficult to directly digest and absorb by the body due to the lack of corresponding catabolic enzymes in humans [6]. However, NPs can exert hypoglycemic functions by regulating gut microbiota, interfering with host metabolism, or directly targeting signaling pathways. For example, some unabsorbed high molecular weight NPs act as prebiotics to regulate the gut microbiota and ferment to produce SCFAs, which activate the secretion of glucagon-like peptide-1 (GLP-1) by enteroendocrine cells. Some low molecular weight NPs that are absorbed in small amounts by the body can enter hepatocytes/adipocytes via endocytosis and target signaling pathways to exert hypoglycemic activity. Matsathit et al. [7] investigated the pancreatic islet-protective effects of split gill mushroom polysaccharides in high-fat diet (HFD) combined with low-dose STZ-induced T2D rats. It was shown that the high-dose polysaccharide group (dose 240 mg/kg) significantly improved islet pathological damage, up-regulated INS levels and Glucose Transporter 2 (GLUT2) protein expression, and reduced pancreatic malondialdehyde (MDA) levels, suggesting that it exerts a hypoglycemic effect through anti-oxidative stress and pancreatic islet β-cell function protection. Some NPs can also enhance the insulin signaling pathway by activating immune receptors. Lu et al. [8] prepared a *Sargassum* seaweed alginate (molecular weight 45.4 kDa), which can alleviate oxidative stress injury by activating nuclear factor E2-related factor 2 (Nrf2). In a diabetic wound model, intravenous injection of sargassum seaweed polysaccharide (200 mg/kg) significantly accelerated wound healing and ameliorated the associated metabolic abnormalities. *Inonotus obliquus* polysaccharide (373 kDa) can repair the gut barrier by up-regulating the expression of Zonula occludens-1 (ZO-1) and Mucin 2 (MUC2) [9].

However, the multi-target synergistic hypoglycemic mechanism of NPs has not yet been fully elucidated, and current research focuses on a single mechanism and lacks the synergistic effect analysis of the “gut microbiota–metabolism–immunity” network, which makes it difficult to predict the superimposed effect of multi-target interventions in the clinical translation. The molecular structure (e.g., molecular weight, branched chain complexity) and biological activity of NPs, as well as the integration mechanism of trans-organ regulatory networks (e.g., intestinal–hepatic and gut microbiota–immune axes), still need to be investigated in depth [10]. In this paper, we systematically review the latest research progress on NPs in terms of conformational relationship, gut microbiota regulation, metabolic regulation, immune regulation and activation of signaling pathways, aiming to provide a theoretical basis for the mechanism analysis, structural optimization and clinical application of NPs.

## 2. Structure–Activity Relationships

### 2.1. Molecular Mass

The molecular weight of NPs is closely related to their biological activities. The higher the molecular weight of NPs, the more obvious the spatial barrier effect; it is difficult for them to directly penetrate the cell membrane or bind to the target, but they can delay glucose absorption through physical adsorption, and indirectly regulate postprandial blood glucose [11]. High molecular weight NPs have poor solubility and are difficult to absorb directly through the gut epithelial cells. For example, konjac glucomannan, with its large molecular weight, dissolves in water to form a high viscosity colloid, which increases the viscosity of gastric contents, prolongs gastric emptying time, enhances satiety, and reduces food intake through volume expansion, indirectly controlling blood glucose fluctuations [12]. In addition, high viscosity polysaccharides can also form a gel-like protective layer in the gut tract, adhering to the surface of the gut mucosa, while wrapping starch granules, preventing starch and digestive enzymes from coming into full contact with each other, thus inhibiting the process of converting starch into glucose [13].

The solubility of medium and low molecular weight polysaccharides is usually better, and they can be transported into the body through the gut mucosa gap or Microfold cell (M cell), and directly act on target organs such as liver and fat. Low molecular weight NPs are more likely to cross the biological barrier and directly inhibit α-glucosidase or α-amylase activity, reducing gut glucose absorption. Lv et al. [14] extracted and purified two polysaccharide fractions, PFP1-1 (52.8 kDa) and PFP2-1 (8.768 kDa), from *Paulownia* flowers. The scavenging ability of PFP2-1 for DPPH, hydroxyl and superoxide anion radicals was significantly better than that of the high molecular weight fraction PFP1-1. The inhibitory activity of PFP2-1 against α-amylase and α-glucosidase was higher, and its high activity might be related to its low molecular weight and glyoxylate structure.

High molecular weight NPs depend on fermentation and catabolism to SCFAs by gut microbiota to indirectly regulate glucose metabolism, e.g., inhibition of hepatic gluconeogenesis and enhancement of INS sensitivity [15]. However, high molecular weight NPs needs to be degraded gradually by the enzyme system of specific microbiota (e.g., *Bacteroides*, *Bifidobacterium*) and slowly release SCFAs, which have the function of regulating glucose homeostasis in the long term. Low molecular weight NPs are more easily degraded by gut microbiota and rapidly fermented to produce SCFAs, which can enhance SCFAs concentration in the short term, but their capacity for sustained regulation is weaker. Lai et al. [16] investigated the hypoglycemic effect of three different molecular weight blackberry polysaccharides in a T2D mouse model. The three polysaccharides were BBP (603.59 kDa), BBP-8 (408.13 kDa) and BBP-24 (247.62 kDa). After eight weeks of intervention, BBP-24 had the best hypoglycemic effect and significantly reduced FBG and area under the curve of the oral glucose tolerance test. All three polysaccharides could increase the abundance of beneficial bacteria (*Mycobacterium* spp.) and inhibit pathogenic bacteria (*Heterobacterium* spp.), among which BBP-24 was more easily fermented and utilized by gut microorganisms due to its lower molecular weight and stronger chain flexibility. Deng et al. [17] compared the hypoglycemic mechanism of two different molecular weight konjac glucans on T2D rats. The results showed that the medium molecular weight polysaccharide KGM-M (757.1 kDa) was superior to the low molecular weight polysaccharide KGM-L (87.3 kDa) in lowering FBG, improving IR and inflammation. KGM-M more significantly enriched the diversity of gut microbiota, elevated the production of SCFAs and the expression of G protein-coupled receptors (GPCRs), and regulated the synthesis of bile acids (BAs), with a significantly higher hypoglycemic activity than that of KGM-L.

Some low molecular weight NPs can enter target cells via transcellular transport or receptor-mediated endocytosis, directly activating intracellular signaling pathways, as well as enter gut epithelial cells via glucose transporters or passive diffusion. For example, some low molecular weight NPs enhance INS sensitivity by activating the Free Fatty Acid Receptor 2 (GPR43)/GLP-1 axis. Some low molecular weight NPs can activate the peroxisome proliferator-activated receptor γ (PPAR-γ) pathway to improve lipid metabolism [18]. Zhao et al. [19] investigated the regulation of GLP-1 secretion by *Lycium barbarum* polysaccharide in STC1 cells and KKAy diabetic mice. *Lycium barbarum* polysaccharide induces rapid GLP-1 release by inhibiting sodium–glucose cotransporter protein 1 (SGLT1) expression, stimulating Ca^2+^ inward flow and inhibiting α-glucosidase activity. Ke et al. [20] established an IR model by tumor necrosis factor-α (TNF-α)/high glucose/INS coinduction to explore the effects of *Astragalus* polysaccharides, which ameliorated the mechanism of IR in 3T3-L1 adipocytes. *Astragalus* polysaccharides activated the PPAR-γ-PI3K/Akt signaling axis by inhibiting microRNA (miR)-721 expression, which, in turn, up-regulated GLUT4 membrane translocation.

Overall, low molecular weight NPs are more readily absorbed by the body, whereas high molecular weight NPs are mainly dependent on gut microbiota metabolism. High molecular weight NPs have significant advantages in immunomodulation and long-lasting glucose regulation through SCFAs, while low molecular weight NPs tend to rapidly affect glucose metabolism by inhibiting digestive enzyme activities or targeting specific signaling pathways.

As shown in Table 1, there are differences in the hypoglycemic mechanism and regulation of related enzyme activities among NPs with different molecular weights. Based on these differences, the synergistic application of polysaccharides of different molecular weights has significant advantages. For example, by mixing polysaccharides of different molecular weights (e.g., inulin with oligofructose), short- and long-term glucose homeostasis can be synergistically regulated. In addition, targeted degradation of high molecular weight NPs by enzymatic (e.g., cellulase, galactosidase) or physical degradation (ultrasound, microwave) techniques can enhance the hypoglycemic activity of NPs. For example, Suo et al. [21] degraded apricot polysaccharides using the dielectric blocking discharge technique combined with the H_2_O_2_-ascorbic acid (VC) Fenton reaction. The degradation process altered the monosaccharide composition of apricot polysaccharides and enhanced their chain linearity. After optimization by response surface methodology, the molecular weight of the degradation product was reduced to 19.71 kDa, and the inhibition rate of α-glucosidase was significantly increased compared with baseline. In the development of hypoglycemic functional products, the molecular weight of NPs should be selected or optimized according to the target mechanism, such as probiotic-mediated regulation of microbiota, inhibition of digestive enzymes, or immune cell signaling.

### 2.2. Branching Complexity

The complexity of the branched chain of NPs is one of the key structural factors affecting their hypoglycemic activity, such as branching density and branching position. The complexity of the branched chain regulates the hypoglycemic effect of NPs by altering their physicochemical properties, interactions with targets, and fermentation characteristics of gut microbiota.

High branching density increases the intermolecular spatial barrier, reduces the formation of hydrogen bonds between molecular chains, enhances water solubility, and facilitates gut dispersion and contact with digestive enzymes. The molecular chains of some low-branching NPs are prone to form tightly ordered helices or crystalline structures, which are poorly water-soluble, but may form a physical barrier through high viscosity to delay glucose release. Konjac glucomannan molecules have very few branching structures in the molecule, and the overall linear backbone is dominant. Its β-1,4 glycosidic bond structure allows it to form a highly viscous gel that wraps around starch granules after water absorption [27].

The branched structure of polysaccharides significantly affects their affinity for biological targets. The branching structure regulates the binding ability of NPs to digestive enzymes and immunoreceptors. A suitable branching structure can expose more active groups (e.g., hydroxyl and carboxyl groups) and enhance the competitive inhibition of α-amylase and α-glucosidase. However, overly complex branching may limit the accessibility of the enzyme due to spatial hindrance and reduce inhibition efficiency. Dendritic cell surface C-type lectin receptors (e.g., Dendritic cell-associated C-type lectin-1 (Dectin-1)) recognize highly branched NPs, triggering endocytosis and activating M2 polarization signals. Glucose residues at the branch ends of highly branched β-(1 → 3)/(1 → 6)-glucans (e.g., *Ganoderma lucidum* polysaccharides) specifically bind to the macrophage Dectin-1 receptor, activate the PI3K/Akt pathway, and enhance insulin signaling [28]. In addition, the branched structure also affects the fermentation properties of NPs. Complexly branched NPs are more likely to be fermented by gut microbiota into SCFAs, which activate GPR43/Free Fatty Acid Receptor 3 (GPR41) receptors, promote GLP-1 secretion, and regulate gut homeostasis [29].

NPs with intermediate branching density and containing functional groups at the end of the branched chain may have better hypoglycemic activity. To optimize the branched structure of NPs, chemical isomerization may be used to increase the diversity of the branched chains or the selective shearing action of specific glycosidases (e.g., cellulase, β-glucanase) may be utilized to obtain the optimal branching density.

### 2.3. Chemical Modification

The solubility, targeting, stability, and ability to interact with biological targets of NPs can be significantly enhanced by introducing specific functional groups or changing the molecular structure, but at the same time may lead to loss of activity or increased toxicity due to over-modification.

Chemical modifications (e.g., sulfation, carboxymethylation, selenylation, acetylation, and phosphorylation) can significantly enhance the hypoglycemic activity of NPs. Xie et al. [30] extracted *Inonotus obliquus* polysaccharides by microwave-assisted extraction and modified them with sulfation, carboxymethylation, phosphorylation, and acetylation. The antioxidant and hypoglycemic activities were evaluated by DPPH/ABTS/hydroxyl radical scavenging capacity, reducing power and α-glucosidase/α-amylase inhibitory activity. The results showed that chemical modifications significantly altered the physicochemical properties of *Inonotus obliquus* polysaccharides, reducing the sugar content, glyoxylate content and molecular weight. Among them, the carboxymethylated derivatives showed the best performance with the lowest molecular weight, higher ABTS radical scavenging rate, stronger reducing power and higher α-amylase inhibition. Fan et al. [31] prepared three derivatives based on *Pueraria lobata* polysaccharides by phosphorylation, carboxymethylation and acetylation modifications. Experiments showed that the chemical modifications significantly altered the physicochemical properties of the polysaccharides, among which the phosphorylation modification led to a substantial increase in the antioxidant activity of *Pueraria lobata* polysaccharides, which was close to that of vitamin C.

The introduction of the sulfate group (-SO_3_H) increases the negative electronegativity of polysaccharides and promotes their binding to positively charged target proteins [32]. For example, sulfated sea cucumber polysaccharides reduced inflammatory state and oxidative stress and improved IR in T2D rats by activating insulin receptor substrate (IRS)/PI3K/Akt signaling and regulating GSK-3β gene expression [33]. Chen et al. [34] sulfated β-glucans (HBGs) from a *Pleurotus tuber-regium* source with different degrees of substitution and evaluated their hypoglycemic and antiglycemic properties to assess their hypoglycemic and antiglycemic activities. The sulfation modifications basically retained the polysaccharide branching structure but reduced the molecular weight, and the introduction of sulfate groups did not disrupt the chain-twisted conformation. S-HBGs showed reversible inhibition of α-glucosidase. In addition, S-HBGs with low to medium substitution were effective in inhibiting protein glycation and the generation of advanced glycation end-products (AGEs).

Carboxymethylation modification significantly enhances water solubility by introducing carboxymethyl (-CH_2_COOH) to break the hydrogen bonds between polysaccharide chains. Carboxymethylated polysaccharides form a viscous gel in the intestine, delaying gastric emptying and glucose absorption, and also enhance the prebiotic properties of the polysaccharides, promoting the proliferation of *bifidobacteria* and increasing the production of SCFAs. Li et al. [35] prepared an activity-enhanced *Undaria pinnatifida* polysaccharide by carboxymethylation modification. The original polysaccharide backbone was dominated by glucose and mannose, whereas carboxymethylation modification introduced glucuronic acids (GalA and GlcA) and localized them at the C-2/C-4 sites. In vivo experiments demonstrated that carboxymethylated polysaccharides significantly enhanced the hypoglycemic, lipid-regulating and antioxidant effects in diabetic mice compared with the original polysaccharides.

The introduction of selenium enhances the antioxidant capacity of polysaccharides and reduces the damage of oxidative stress on pancreatic β-cells. Gao et al. [36] compared the interventional effects of three selenated green tea polysaccharides on T2D. The synthetic selenated polysaccharides significantly improved glucose metabolism and INS sensitivity through activation of the PI3K/Akt pathway, while reducing hepatic oxidative stress and inflammatory responses. The experiments showed that the synthetic selenated polysaccharides were superior to natural and hybrid selenopolysaccharides in metabolic regulation, anti-inflammation and microecological modulation.

The hydrophobicity of acetyl (-OCOCH_3_) promotes the penetration of polysaccharides into cell membranes and their direct action on intracellular targets, which improves the aqueous solubility and bioavailability of NPs. Omer et al. [37] studied the hypoglycemic activity of acetylmannan (Acemannan) from Aloe vera through an STZ-induced diabetic rat model. It was confirmed that Acemannan could significantly improve disorders of glycolipid metabolism, inhibit oxidative damage and inflammation, and restore hepatic glycogen reserves. He et al. [38] found that acetylation modification significantly improved the solubility and surface porosity of an *Houttuynia cordata* rhizome polysaccharide, and increased its inhibitory effect on α-glucosidase.

In addition, complex modifications can better enhance the hypoglycemic activity of NPs; for example, multiple modifications of NPs, such as combined sulfation and carboxymethylation, can better balance solubility, targeting, and safety. Tao et al. [39] prepared selenated apple pectin with a molecular weight of 8.91 kDa by enzymatic modification combined with selenylation, with selenium substitution at the C-6 position of pectin. It significantly reduced FBG, improved glucose tolerance and IR, and attenuated liver/kidney/pancreas injury in the HFD/STZ-induced T2D mouse model.

However, excessive modification may disrupt the active conformation of the polysaccharide, and residues of certain modifying reagents may cause cytotoxicity, such as chlorosulfonic acid for sulfation. Highly modified polysaccharides may be difficult to degrade for gut microbiota and lose their prebiotic function. Therefore, during the modification process, safety assessment of NPs is needed to systematically examine the acute toxicity, long-term metabolism, and gut microbiota effects of the modification products.

## 3. Regulation of Gut Microbiota

The gut microbiota is a complex ecosystem composed of major groups of bacteria such as *Firmicutes*, *Bacteroidetes*, *Proteobacteria* and *Actinobacteria*, which is known as the “second genome of the human body”. It is involved in key physiological processes such as glucose and lipid metabolism, immune regulation and metabolite synthesis [40].

Disturbances of the gut microbiota are strongly associated with T2D, and microbiota remodeling has emerged as a new strategy for diabetes treatment. The gut microbiota of patients with T2D is characterized by a reduced diversity of organisms, an imbalance in the ratio of thick-walled phylum/Anthrobacterium and a significant reduction in the microbiota of SCFAs-producing organisms (e.g., *Roseburia intestinalis*, *Faecalibacterium prausnitzii*) [41]. Probiotics, prebiotics and other gut microbiota regulators can regulate the imbalance of the microbiota and improve the levels of glycosylated hemoglobin and fasting INS. NPs (e.g., algal polysaccharides) are not well-absorbed in the body’s upper digestive tract after being ingested, whereas once in the gut environment, they can become essential nutrients for the gut microbiota [42]. NPs act as a prebiotic to provide a carbon source for the probiotic bacteria, and at the same time, probiotics metabolize the NPs to generate SCFAs and enhance the gut function of the microbial microbiota. SCFAs are generated by probiotics to enhance the gut barrier and GLP-1 secretion [43].

SCFAs protect the gut barrier and regulate glucose metabolism through multiple pathways such as decreasing gut permeability and inhibiting the production of pro-inflammatory factors [44]. SCFAs also reduce endogenous glucose production by directly promoting hepatic glycogen synthesis and inhibiting the process of gluconeogenesis through the AMPK pathway. For example, butyric acid can reduce hepatic gluconeogenesis by inhibiting the activity of histone deacetylases (HDACs), up-regulating the expression of pancreatic developmental genes such as Pancreatic and Duodenal Homeobox 1 (PDX-1), promoting the secretion of INS, and enhancing mitochondrial β-oxidation. Propionic acid directly inhibits lipolysis and reduces free fatty acid (FFA) levels, and also reduces hepatic gluconeogenesis by down-regulating the expression of PEPCK [45]. *Artemisia argyi* polysaccharide increased the abundance of SCFAs, especially butyric acid, in the gut contents and feces of diabetic mice, and increased the abundance of *Allobaculum* spp. in the cecum [46]. Zhou et al. [47] investigated the regulatory effects of fucoidan on human gut microbiota and their metabolites using an in vitro fermentation model. Fucoidan increased total SCFAs, acetic acid and propionic acid, significantly reduced the abundance of pathogenic *Escherichia-Shigella* and *Klebsiella*, and promoted the probiotics *Bifidobacterium* and *Klebsiella*.

Some NPs can exert hypoglycemic effects by increasing the abundance of beneficial bacteria such as *Ackermannia*, thick-walled bacteria and *Lactobacillus*, decreasing the abundance of harmful bacteria such as *Enterobacteriaceae*, *Clostridium perfringens* and *Vibrio desulfuricans*, regulating the disorder of the gut microbiota, improving the glycolipid metabolism, and increasing the content of SCFAs [48]. *Polygonatum sibiricum* polysaccharide significantly improved the T2D rats’ metabolic abnormalities, decreased the ratio of gut microbiota thick-walled phyla/*Anthrobacterium* phyla, up-regulated the beneficial bacteria such as *Blautia* and *Ackermannia*, and down-regulated the harmful bacteria such as *Prevotella* and *Escherichia coli* [49]. Tong et al. [50] investigated the hypoglycemic effect of *Laminaria japonica* polysaccharides (LJP) in T2D model mice. After 4 weeks of continuous gavage, LJP significantly reduced FBG and INS concentrations and improved histopathological damage. LJP also increased probiotic abundance, increased the anabolic phylum/thick-walled phylum ratio, inhibited the growth of harmful bacteria, and promoted the proliferation of SCFAs-producing bacteria, which elevated the level of cecal SCFAs.

### 3.1. Gut–Liver Axis Regulatory Network

The gut microbiota affects gut barrier function, body immunity and glycolipid metabolism. Disturbances in the gut microbiota lead to a decrease in SCFAs, which triggers abnormal metabolism of BAs, branched-chain amino acids (BCAAs), and lipopolysaccharides (LPS), which in turn leads to T2D through IR and chronic inflammation [51]. Dysbiotic microbiota also alter the structure of BAs, which impairs their ability to regulate glucose and lipid metabolism through the Farnesoid X receptor (FXR)/G-protein-coupled receptor 5 (TGR5) receptor to regulate glycolipid metabolism [52].

Some NPs can activate TGR-5 receptors and enhance GLP-1 secretion by enriching *Clostridium perfringens*, up-regulating bile salt hydrolase (BSH) activity, and promoting the conversion of primary to secondary BAs. BSH activation also reduces hepatic gluconeogenesis by inhibiting FXR signaling. BAs can promote glycogen synthesis through the AMPK signaling pathway and optimize the balance of glucose–lipid metabolism at the liver level. Dai et al. [53] studied the hypoglycemic effects and regulatory mechanisms of the hypoglycemic effect and regulatory mechanism of mulberry leaf polysaccharides (MLP) in T2D rats. MLP significantly reduced FBG/lipid levels, ameliorated disorders of glucose–lipid metabolism and IR, and remodeled gut microbiota homeostasis by enriching *Prevotella*, *Ruminococcus* and *Lactobacillus*. MLP activated the classical BAs synthesis pathway, down-regulated hepatic and ileal FXR expression, up-regulated cholesterol 7α-hydroxylase (CYP7A1) and 12α-hydroxylase (CYP8B1) expression, promoted the conversion of cholesterol to BAs, and activated ileal TGR5 to synergize with the improvement of glycolipid metabolism.

SCFAs also activate GPCRs, such as free fatty acid receptors GPR43 (FFAR2) and GPR41 (FFAR3), and downstream signaling pathways such as AMPK and PI3K to exert hypoglycemic effects. When GPR41/43 is activated, it promotes the secretion of GLP-1 from gut L cells [54]. Liu et al. [55] investigated the preventive effect and hypoglycemic mechanism of *Dendrobium officinale* polysaccharide (DOP) in pre-diabetic mice. DOP was found to be able to repair islet damage, suppress appetite and improve glucose metabolism. A DOP dose of 200 mg/kg/d reduced the relative risk of prediabetes to T2D by 63.7%. DOP regulated the composition of gut microbiota, reduced LPS levels and inhibited the expression of the TLR4 inflammatory pathway to alleviate IR. DOP also enriched the SCFAs-producing microbiota and elevated the content of SCFAs in the intestine, activating the SCFAs receptors FFAR2/FFAR3 to promote the secretion of the gut hormones GLP-1 and Peptide YY (PYY). Chayote pectin can synergistically regulate blood glucose through three pathways: “gut microbiota/SCFAs/GLP-1”, “PI3K/Akt/FoxO1” and “GPR43/AMPK/FoxO1” [56]. The specific hypoglycemic mechanism of Chayote pectin is mainly manifested in four ways: first, through the activation of GPR43 and P-AMPK; second, through the remodeling of the gut microbiota structure by increasing the abundance of beneficial bacteria and decreasing the ratio of thick-walled bacillus/anabolic bacillus; third, it significantly up-regulated the expression of PI3K and phosphorylated Akt (p-Akt), while down-regulating the expression of FoxO1, G6Pase and PEPCK; fourth, the expression of tight junction protein Claudin-1 and mucin MUC2 was significantly up-regulated to improve the gut barrier function.

The Cyclic Adenosine Monophosphate (cAMP)/Protein Kinase A (PKA) signaling pathway is the core network of hormone-regulated glucose metabolism and plays an important role by regulating INS secretion, glycogenolysis, and gluconeogenesis [57]. GLP-1 triggers the Gαs-adenylate cyclase (AC)-cAMP cascade by activating the β-cell membrane receptor, GLP-1R, to elevate intracellular cAMP levels, which is then synergistically promoted by PKA and Epac2 to enhance INS secretion. INS secretion is promoted by PKA, which phosphorylates cAMP response element binding protein (CREB) and enhances insulin gene transcription. Epac2 closes K^+^-ATP channels, activates voltage-dependent calcium channels (VDCC), and triggers calcium ion (Ca^2+^) influx and insulin vesicle exocytosis, thereby promoting INS release [58]. In addition, the cAMP/PKA pathway activates glycogen phosphorylase (GP), which promotes glycogenolysis, as well as inhibits PEPCK expression [59].

GLP-1 enhances INS release from β-cells via the AC/cAMP/PKA axis. GLP-1 is also an enteroglucagon that inhibits glucagon secretion and delays gastric emptying. Liu et al. [60] investigated the mechanism of action of a neutral polysaccharide (mannogalactan glucan) from *Armillaria mellea* to improve pancreatic islet function in type 1 diabetes (T1D). Mannogalactan glucan promotes Ca^2+^ endocytosis and triggers the transport of insulin-secreting vesicles by activating the Calcium/calmodulin-dependent protein kinase II (CamkII) and cAMP/PKA pathways. Xia et al. [61] systematized the effects of *Achyranthes bidentata* polysaccharide (ABP) on insulin secretion through a multicascade mechanism based on the “colony–metabolism–signaling pathway”. ABP repairs the gut barrier and enhances GLP-1 secretion by enriching SCFAs-producing bacteria, such as *Roseburia* and *Faecalibacterium*, and promotes INS synthesis and release through activation of the GLP-1R/cAMP/PKA/CREB signaling axis. The ABP-mediated colony–SCFAs–GLP-1 metabolic loop forms a positive regulatory network with host INS signaling, which may provide a theoretical basis for the development of polysaccharide-based antidiabetic drugs targeting the “gut–liver–islet axis”.

### 3.2. Gut Microbiota–Immunity Axis Regulatory Network

The dysbiosis of gut microbiota can increase gut permeability by disrupting the integrity of the gut mucosal barrier, leading to the entry of LPS and FFA into the bloodstream. LPS/FFA induces the release of pro-inflammatory cytokines mainly through binding to TLR4, recruiting myeloid differentiation factor 88 (MyD88), and activating the NF-κB and mitogen-activated protein kinase (MAPK) signaling pathways. Pro-inflammatory cytokines such as TNF-α and interleukin-6 (IL-6) can lead to impaired insulin signaling by interfering with the IRS phosphorylation cascade.

Some NPs block TLR4 binding to LPS/FFA, inhibit MyD88-dependent NF-κB nuclear translocation, down-regulate pro-inflammatory factors such as TNF-α and IL-6, and attenuate β-cell inflammatory injury. Cordyceps polysaccharide (156.511 kDa) inhibits the release of LPS into the bloodstream by decreasing the abundance of gut Gram-negative bacteria, blocking the TLR4/MyD88/NF-κB inflammatory pathway in liver and adipose tissue, and inhibiting the LPS/TLR4 inflammatory pathway in liver, thus improving IR [62]. Zhang et al. [63] demonstrated that acetylated *Ganoderma applanatum* polysaccharides (AGAP) improve the T2D through multi-target regulation of AGAP-inhibited reactive oxygen species (ROS) accumulation by activating the Nrf2/Keap1 antioxidant pathway, and blocked the TLR4/NF-κB inflammatory cascade, while regulating Bax/B-cell lymphoma-2 protein (Bcl-2) balance to reduce apoptosis. In addition, AGAP maintained the integrity of the gut barrier, increased the level of gut tight junction proteins, and effectively prevented the metastasis of LPS to the liver.

Some NPs can regulate blood glucose through the microbiota–immunity axis, and improve metabolic disorders by regulating gut microecology, inhibiting inflammation, and enhancing antioxidant capacity through multiple pathways. *Grifola frondosa* polysaccharide significantly alleviated HFD-induced IR, improved gut microbiota disorders, and increased the production of SCFAs, which in turn inhibited the LPS/TLR4 inflammatory pathway. At the same time, it activated the Nrf2/ARE antioxidant signaling pathway and attenuated oxidative stress [64]. Liu et al. [65] investigated the regulatory effects of *Dioscorea opposita* polysaccharides (DOP) on IR, lipid metabolism, oxidative stress, and gut microbiota of HFD combined with STZ-induced T2D rats. A high dose of DOP (400 mg/kg) significantly improved glucose–lipid metabolism and hepatic oxidative stress. DOP increased the levels of GLP-1, High-density lipoprotein cholesterol (HDL-C), alleviated hepatic oxidative damage and reduced hepatic steatosis by elevating the levels of superoxide dismutase (SOD), catalase (CAT), and glutathione (GSH), and by lowering the level of MDA. Qiao et al. [66] investigated the hypoglycemic effect of *Schisandra chinensis* acidic polysaccharide (SCAP) by establishing a T2D rat model by HFD combined with STZ (30 mg/kg). SCAP was shown to inhibit the activation of the c-Jun N-terminal kinase (JNK)/NF-κB pathway, reduce the release of pro-inflammatory factors, and block inflammation-mediated impairment of insulin signaling. SCAP enhances the phosphorylation level of the insulin receptor substrate-1 (IRS-1)/PI3K/Akt pathway, restores signaling downstream of the INS receptor, and promotes glucose uptake and utilization.

Some NPs may alleviate chronic low-grade inflammation by increasing tight junction protein expression and reducing LPS entry into the bloodstream, e.g., ZO-1, Occludin, etc. [67]. Some NPs may prevent pathogenic bacterial invasion by promoting mucin MUC2 secretion and maintaining mucus layer thickness. *Fructus mori* polysaccharide alleviates hyperglycemia, IR, hyperlipidemia, and endotoxemia levels in T2D mice. It repaired the gut barrier by inhibiting the activation of the TLR4/MyD88/NF-κB pathway, suppressing the level of gut inflammation and oxidative stress, and indirectly up-regulating the expression of the tight junction proteins, Claudin-1, Occludin, and ZO-1 [68]. Siddiqui et al. [69] investigated the effects of crude *Dictyopteris divaricata* polysaccharide (CDDP) amelioration in STZ-induced T1D-Balb/c mice. The results showed that CDDP intervention significantly improved FBG, oral glucose tolerance (OGTT), serum INS levels, modulated the levels of pro-inflammatory factors, and increased the abundance of beneficial bacteria, such as thick-walled bacteria, *Mycobacterium avium*, and *Lactobacillus* spp. CDDP attenuated colon inflammation and repaired the gut barrier structure and permeability by up-regulating the expression of IRS-1, MUC2, and tight junction protein (TJ). This study provides experimental evidence for the synergistic alleviation of T1D by NPs through the multi-targeting mechanism of “microbiota–gut barrier–islet regeneration”.

In addition, some NPs reduce pro-inflammatory factors by blocking Inhibitor of nuclear factor kappa B alpha (IκBα) degradation and maintaining the NF-κB p65 subunit retention in the cytoplasm. SCFAs also inhibit nuclear translocation of the NF-κB pathway through activation of the GPR41/43 receptor, reducing pro-inflammatory factors. Sweet corn cob selenium polysaccharide decreased gut barrier permeability, modulated gut microbiota structure, increased *Firmicutes* abundance, and inhibited the LPS/IKBα/NF-κB inflammatory pathway to reduce inflammatory factor production, thereby enhancing glycolipid metabolism [70]. Song et al. [71] found that *Astragalus membranaceus* polysaccharides (AMP) ameliorated the mechanism of diabetes in db/db mice by regulating the gut microbiota. Oral administration of AMP (600 mg/kg/d × 16 d) significantly alleviated the symptoms of hyperglycemia, increased the production of gut SCFAs and remodeled the structure of gut microbiota, and significantly increased the abundance of beneficial bacteria, such as *Akkermansia*, *Faecalibaculum*, etc. AMP promoted the secretion of serum GLP-1 through the activation of GPR41/43 by SCFAs and up-regulated the secretion of tight junction proteins (e.g., Occludin, ZO-1) expression to enhance the gut barrier function.

Overall, NPs reduce the penetration of harmful substances by enhancing gut barrier integrity, while inhibiting the TLR4/MyD88/NF-κB pathway to reduce inflammatory factor release. Probiotics metabolize NPs to generate SCFAs, which can regulate blood glucose and improve insulin resistance through various pathways, and NPs can also regulate lipid metabolism and gluconeogenesis through BSH/FXR targets to synergistically exert hypoglycemic functions (Figure 1).

## 4. Regulation of Metabolism

Glucose metabolism involves the process of glucose decomposition, synthesis and storage in the body, including glycolysis, gluconeogenesis, glycogen synthesis, glycogenolysis and other processes. The liver plays an important role in the process of glucose metabolism, which is not only the main site of glycogen synthesis and catabolism, but also a key organ for gluconeogenesis. The occurrence of IR in liver tissue will lead to hyperglycemia and decreased glucose tolerance, which will lead to the development of T2D. Some NPs can exert hypoglycemic activity by promoting the processes of glycolysis and glycogen synthesis, inhibiting glycogenolysis and gluconeogenesis.

### 4.1. Enhanced Insulin Signaling

Impairment of insulin transduction signaling induces the formation of IR, such as signaling disorders of INSR, IRS, PI3K, Akt, etc. INSR is a receptor protein for INS, located on the membrane of the target cell where INS acts, and binding to INS opens up a transporter pathway on the cell membrane, whereby glucose enters into the cell with the assistance of the transporter [72]. Weakening of the response of INSR to INS will lead to a decrease in the ability of muscle and adipose tissue to take up glucose, which can lead to diabetes and its complications. IRS is a signaling molecule, and its expression level directly affects insulin signaling. Aberrant expression or dephosphorylation of INSR and IRS impedes insulin signaling, and protein tyrosine phosphatase 1B (PTP1B) promotes the dephosphorylation of both, which directly inhibits the insulin signaling pathway [73]. Some NPs can increase the expression level of GLUT4 by inhibiting the mRNA expression level of PTP1B, thereby increasing INS sensitivity and improving IR [74]. Some NPs can also promote the phosphorylation of INSR and IRS, activate downstream PI3K/Akt signaling, and enhance the membrane translocation of GLUT4 to promote the glucose uptake of peripheral tissues [75]. Yu et al. [76] found that *Ganoderma lucidum* proteoglycan significantly enhanced glucose uptake in IR-HepG2 cells and potently inhibited the activity of PTP1B.

The PI3K/Akt pathway is the core pathway of insulin signaling, which achieves hypoglycemic effects by promoting glycogen synthesis, glucose transport and inhibiting gluconeogenesis [77]. Its core functions include the following: promoting the translocation of GLUT4 to the cell membrane to enhance glucose uptake in muscle and adipose tissue; inhibiting GSK-3β and activating GS to promote hepatic glycogen storage; regulating lipid metabolism, anti-apoptosis and anti-inflammatory responses; and protecting pancreatic islet β-cells [78].

PI3K consists of a regulatory subunit (e.g., p85) and a catalytic subunit (e.g., p110). Binding of INS to the receptor activates PI3K, which catalyzes the generation of the second messenger PIP3 from the membrane phospholipid PIP2. PIP3 recruits and activates Akt, which further phosphorylates downstream target proteins. p-Akt increases cellular glucose uptake by facilitating the translocation of GLUT4 to the plasma membrane [79]. p-Akt also enhances GS function and promotes hepatic glycogen storage by phosphorylating the Ser9 site of GSK-3β and inhibiting its activity [80]. In addition, p-Akt reduces oxidative stress-induced β-cell apoptosis by up-regulating Bcl-2 and inhibiting caspase-3 activity. p-Akt activation also inhibits NF-κB inflammatory signaling and reduces the release of inflammatory factors, such as IL-6 and TNF-α, which indirectly protects the function of β-cells [81].

NPs activate the PI3K/Akt pathway downstream of INSR, induce GS phosphorylation and enhance its activity, and inhibit GP expression, thereby increasing hepatic glycogen storage [82]. For example, Zhang et al. [83] investigated the molecular mechanism by which black tea polysaccharides (BTP) improve T2D based on the “PI3K/Akt/GLUT2” signaling axis. Chemical characterization showed that its highly branched structure, dominated by glucuronides, may enhance hypoglycemic activity. Oral administration of BTP dose-dependently reduced FBG in T2D mice by 37.88% in the 800 mg/kg dose group compared to diabetic controls. BTP enhances INS sensitivity by phosphorylating the activated PI3K/Akt pathway and promotes the translocation of GLUT2 membranes to accelerate glucose uptake in hepatocytes. Zhang et al. [84] investigated the molecular mechanisms underlying the improvement of glucose uptake in *Cynanchum auriculatum* Royle ex Wight Polysaccharides (CRP) on hyperglycemia and gut microbiota in T2D mice. CRP exerted hypoglycemic effects by activating the IRS-1/PI3K/Akt-1/GLUT2 signaling pathway, which enhanced the mRNA expression of IRS-1, PI3K, Akt-1, and GLUT2 and enhanced INS sensitivity and glucose transport. CRP also modulates the structure of gut microbiota, enriching SCFAs-producing Limosillactobacillus and anti-inflammatory-associated Prevotella.

Glucose Transporter (GLUT) is one of the most important signaling molecules downstream of phosphorylation of INS transduction-related signaling pathways in liver, adipose and skeletal muscle tissue [85]. When blood glucose level increases, glucose enters β-cells via GLUT2 and undergoes oxidative phosphorylation to generate ATP, which closes the K^+^-ATP channel and activates the VDCC, and ultimately triggers INS secretion. The secreted INS binds to the INSR on the membrane surface of target cells, activates downstream signaling pathways such as PI3K/Akt and MAPK, and promotes the translocation of GLUT4 to the cell membrane, accelerating glucose uptake and metabolism in peripheral tissues [86]. Increased expression of GLUT4 increases the cellular sensitivity to INS, which promotes glucose entry into the cell. Sun et al. [87] performed low-temperature plasma modification of apricot pectic polysaccharide. The modified polysaccharide increased the ADP/ATP ratio, inhibited PKA phosphorylation, and also activated the AMPK-peroxisome proliferator-activated receptor γ coactivator-1α (PGC-1α) pathway, which stimulated mitochondrion generation and facilitated GLUT4 protein translocation. Coix seed polysaccharide (7.75 kDa) can promote glucose entry into cells by activating the PI3K/Akt signaling pathway to up-regulate GLUT4 expression and significantly enhance glucose uptake in IR-HepG2 cells, which has the potential to improve the prognosis in IR and related metabolic disorders [88].

### 4.2. Improvement of Insulin Resistance

When cellular sensitivity to INS is reduced, glucose uptake and utilization are impaired, leading to IR. IR formation stems from a multimechanism regulatory imbalance involving four main aspects: first, reduced tyrosine phosphorylation of IRS proteins, which impairs GLUT4 translocation, leading to reduced glucose uptake and impaired muscle/adipose tissue glucose utilization. Second, lipotoxic substances such as ceramides phosphorylate IRS-1 serine through the Protein Kinase C theta (PKCθ)/JNK pathway, resulting in negative feedback inhibition, thus blocking IRS-1 binding to PI3K and inhibiting insulin signaling. Third, chronic inflammation activates the JNK-Signal Transducer and Activator of Transcription 3 (STAT3) axis, inducing Suppressor of Cytokine Signaling 3 (SOCS3) expression and blocking IRS-1 signaling. Fourth, reduced mitochondrial ATP production in skeletal muscle impairs glucose oxidation, which leads to decreased skeletal muscle glucose metabolism and FFA accumulation [89].

NPs can improve IR by regulating the above mechanisms. *Dendrobium officinale* polysaccharides significantly improved dysglycemia, hepatic oxidative stress, and inflammatory infiltration in obese rats by down-regulating SOCS3 expression in the Janus Kinase (JAK)/STAT3/SOCS3 signaling pathway and up-regulating IRS-1. The polysaccharide intervention increased the abundance of beneficial gut microbiota, and the changes in the microbiota were significantly correlated with IRS-1 expression and inflammatory factors [90]. Zhang et al. [91] found that *Sargassum fusiforme* fucoidan significantly elevated Tauroursodeoxycholic Acid (TUDCA) levels in the colon, inhibited the FXR signaling pathway, and ameliorated IR in diet-induced obese mice. *Sargassum fusiforme* fucoidan reduced ceramide levels in serum and colonic tissues and inhibited FXR-regulated ceramide synthesis key enzyme gene expression in mice.

The MAPK signaling pathway is the core regulatory network of cells in response to external stimuli (e.g., inflammation, oxidative stress, hormonal signals) and plays an important role in glucose metabolism, INS sensitivity, and pancreatic β-cell function. The MAPK signaling pathway is a core regulatory network consisting of a highly conserved three-tiered kinase cascade (MAP3K → MAP2K → MAPK), whose subfamilies include extracellular signaling regulatory kinase (ERK), JNK, and p38 MAPK. ERK signaling promotes β-cell proliferative gene expression and inhibits apoptotic protein activity by phosphorylating transcription factors. However, the ERK pathway has a dual role in INS secretion; low concentrations of glucose promote β-cell proliferation through ERK1/2 activation, but its over-activation may inhibit IRS-1 tyrosine phosphorylation and exacerbate IR [92]. Over-activation of JNK leads to serine phosphorylation of IRS-1, blocks its tyrosine phosphorylation, inhibits PI3K/Akt signaling, and also activates c-Jun/Activator Protein 1 (AP-1) transcription factor, which induces apoptosis in pancreatic β-cells. p38 MAPK reduces INS secretion by inhibiting Phosphoinositide-dependent kinase 1 (PDK1) phosphorylation, activates NF-κB and NOD-like receptor protein 3 (NLRP3) inflammatory vesicles, promoting the release of pro-inflammatory factors and exacerbating islet inflammatory injury [93].

Some NPs can improve IR by regulating key nodes of the MAPK pathway, such as ERK, JNK, etc. Pan et al. [94] investigated the protective and reparative effects of *Ganoderma lucidum* proteoglycan (GLP) on oxidative stress-injured pancreatic β-cells. GLP possesses antioxidant activities such as free radical scavenging and reducing power. GLP improved H_2_O_2_-induced INS-1 cell viability, INS secretion, and mitochondrial membrane potential through the regulation of the apoptosis-associated MAPK/NF-κB pathway and INS secretion pathway. GLP significantly repaired pancreatic injury, reduced β-cell apoptosis, and regulated the redox homeostasis of db/db diabetic mice, and thus improved INS secretion.

### 4.3. Inhibition of Sugar Digestion

Sugars in food are first hydrolyzed to oligosaccharides (e.g., maltose) by pancreatic α-amylase in the small intestine and subsequently broken down by α-glucosidase (e.g., maltase) in the brush border of the small gut mucosa to glucose, which is absorbed through the intestine leading to elevated blood glucose [95]. When the activity of these two enzymes is abnormally elevated, carbohydrates are rapidly broken down into glucose and absorbed into the bloodstream, triggering dramatic fluctuations in postprandial blood glucose. Therefore, α-amylase and α-glucosidase are key targets for diabetes drugs such as acarbose. Some NPs retard carbohydrate catabolism and glucose uptake by competitively binding or non-competitively inhibiting the activities of these two enzymes. Zhang et al. [96] investigated the hypoglycemic mechanism of *Panax notoginseng* leaf polysaccharide (16.57 kDa) and found that it could be used to inhibit α-glucosidase and α-amylase activities. Ni et al. [97] found that *Cordyceps militaris* polysaccharides (CM-P) could ameliorate diabetic complications by modulating metabolic and antioxidant/anti-inflammatory pathways. CM-P was found to significantly inhibit α-amylase and α-glucosidase activities and exhibited multidimensional hypoglycemic effects in diabetic mice. CM-P lowered blood glucose and INS levels, enhanced insulin sensitivity, improved disorders of glucose–lipid metabolism, alleviated oxidative stress and inflammatory responses, and repaired diabetes-related organ damage.

### 4.4. Regulation of Key Enzyme Activities

As shown in Table 2, in the regulation of hepatic glucose metabolism, GK, G6Pase, GS, and GP are the four core enzymes, which control glucose uptake, glucose output, glycogen synthesis, and glycogenolysis, respectively, and constitute the core regulatory axes of glucose homeostasis [98]. In addition, hexokinase (HK), GSK-3β and pyruvate kinase (PK) are also important enzymes in the regulation of glucose metabolism. HK is a key enzyme in the first step of the reaction of glycolysis, PK plays a role in the last step of glycolysis and accelerates glycolysis, and GSK-3β is a key enzyme in the blocking of glycogen synthesis.

When blood glucose is elevated, INS secretion increases in the organism, and INS activates GS to promote glucose synthesis of hepatic glycogen, inhibits GP and G6Pase activity, and reduces hepatic glycogenolysis and gluconeogenesis, thus lowering blood glucose levels [99]. On the contrary, INS deficiency triggers glucose metabolism disorders. On the one hand, INS deficiency reduces the activity of GK, which in turn inhibits hepatic uptake and utilization of glucose, leading to reduced glucose consumption. On the other hand, INS also enhances the activity of GP and G6Pase, which promotes hepatic glycogenolysis and gluconeogenesis, ultimately resulting in increased FBG levels.

Some NPs are able to maintain blood glucose stability by regulating the activity of these key enzymes of glucose metabolism in the liver. For example, Liu et al. [100] developed an oral solution containing *Magnolia officinalis* polysaccharide and explored its hypoglycemic effect. In an STZ-induced diabetic mouse model, after 8 weeks of gavage in the high-dose group (10 mL/kg), mice showed a significant reduction in blood glucose, improvement in glucose tolerance, elevated fasting INS levels, reduction in dyslipidemia and oxidative stress, and a significant enhancement in HK and PK activities. The study showed that the oral solution exerted antidiabetic effects by promoting glucose utilization in peripheral tissues, elevating INS levels, enhancing antioxidant capacity and the activities of key enzymes of glycolysis. Sun et al. [101] investigated the effects of tomato pectin on HFD-induced hepatic IR and its potential mechanisms. Tomato pectin significantly reduced liver weight, fat accumulation and liver injury, and improved FBG and glucose tolerance in HFD mice. Tomato pectin up-regulates GLUT4 expression and inhibits key enzymes of gluconeogenesis (PECK, G6Pase) by modulating the PI3K/Akt/GSK-3β signaling pathway and restores antioxidant activity and inhibits hepatic inflammatory state, thereby improving IR.

In addition, some NPs can inhibit the expression of key enzymes of gluconeogenesis by activating the AMPK pathway, while promoting GLUT4 translocation and improving IR. AMPK is a central regulator of cellular energy metabolism. AMPK is a heterotrimer consisting of α (catalytic subunit), β (scaffolding subunit), and γ (regulator subunit). Phosphorylation of the α-subunit at the Thr172 site is a key marker of its activation. AMPK phosphorylation is triggered when the cellular energy state is altered or when exogenous stimuli pass through upstream kinases, such as Liver Kinase B1 (LKB1), or Calmodulin-dependent protein kinase β (CaMKKβ).

Activated AMPK inhibits gene transcription of G6Pase and PEPCK by phosphorylating the transcription coactivators CREB-regulated transcription coactivator 2 (CRTC2) and FoxO1, and also up-regulates the expression of antioxidant enzymes, such as SOD and CAT, to attenuate the damage to β-cells by ROS. AMPK can also activate the PI3K/Akt/GSK-3β signaling pathway, enhance GS activity, and increase hepatic glycogen synthesis in the organism.

At the level of lipid metabolism regulation, AMPK promotes fatty acid β-oxidation and reduces lipid accumulation by inhibiting Acetyl-CoA Carboxylase (ACC) and activating Carnitine Palmitoyltransferase 1 (CPT-1). Some NPs also enhance PGC-1α activity and promote mitochondrial biogenesis and fatty acid oxidation-related gene expression by activating Sirtuin 1 (Sirt1)-mediated deacetylation. Some NPs inhibit the activity of key transcription factors for adipocyte differentiation (e.g., PPAR-γ), reduce triglyceride synthesis, and promote lipolysis and reduce intracellular lipid accumulation through up-regulation of Adipose Triglyceride Lipase (ATGL) and Hormone-Sensitive Lipase (HSL). Chen et al. [102] verified the protective effect of Tieguanyin oolong tea polysaccharides (TPS) in an HFD-induced NAFLD mouse model. TPS up-regulates the expression of key enzymes including Sirt1, ACC1 and ATGL by activating the AMPK signaling pathway. Experiments have shown that TPS significantly ameliorates hepatic lipid metabolism disorders, glucose intolerance, and steatosis. Wang et al. [103] investigated the hypoglycemic mechanism of sweet corncob polysaccharide in T2D mice, and found that it significantly reduced blood glucose and lipid deposition. Sweet corncob polysaccharide can regulate the AMPK/ACC/CPT-1 signaling pathway in the liver, as well as the AMPK/Sirt1/PGC-1α and AMPK/Sirt1/Uncoupling Protein 2 (UCP2) pathways in the pancreas, which can effectively improve disorders of glucose and lipid metabolism, and enhance the function of pancreatic islet β-cells.

### 4.5. Inhibition of Gluconeogenesis

FoxO1 is a key protein in the regulation of gluconeogenesis, which is located downstream of the insulin signaling pathway, and the level of phosphorylated forkhead box protein (p-FoxO1) plays a decisive role in the transcriptional level of the gluconeogenesis-regulating enzymes. G6Pase is the active enzyme that regulates the last step of gluconeogenesis and PEPCK is the active enzyme that catalyzes the first step of gluconeogenesis, and the transcript levels of both are regulated by the upstream factor FoxO1. In the diabetic state, gluconeogenesis is enhanced, and the activities of PEPCK and G6Pase are elevated, which will lead to blood glucose elevation [104]. PGC-1α, as the core regulatory hub of gluconeogenesis, directly activates the expression of G6Pase, PEPCK and other genes by binding to transcription factors such as FoxO1, CREB and so on.

In addition, fructose-1,6-bisphosphatase (FBPase) and pyruvate carboxylase (PC) are rate-limiting enzymes for gluconeogenesis. FBPase catalyzes the gluconeogenesis branch point reaction, hydrolyzes FBP-1 to fructose-6-phosphate, and its activity is regulated by the AMP/ATP ratio [105]. Activation of the AMPK pathway induces inactivation of FBPase phosphorylation, thereby inhibiting excessive gluconeogenesis. PC connects glycolysis and gluconeogenesis and catalyzes the generation of oxaloacetate from pyruvate, the expression of which is affected by the NF-κB inflammatory pathway.

Some NPs reduce the expression of gluconeogenesis-limiting enzymes (G6Pase, PEPCK) by inhibiting the phosphorylation of FoxO1 and its binding to PGC-1α. Xu et al. [106] found that *Astragalus* polysaccharide could ameliorate diabetic nephropathy by targeting the Sirt1/FoxO1 autophagy pathway. Experiments showed that *Astragalus* polysaccharide significantly reduced renal histopathological injury, oxidative stress, and inflammatory responses in rats. Luo et al. [107] investigated the molecular mechanism of *Pueraria lobata* root polysaccharide (PLP) to ameliorate diabetic metabolic syndrome by targeting the PI3K/Akt signaling pathway. PLP was shown to significantly reduce body weight, visceral fat accumulation, and FBG and IR index in db/db mice. PLP can effectively improve lipid metabolism disorders and reduce Triglyceride (TG), Total Cholesterol (TC), Low-Density Lipoprotein Cholesterol (LDL-C) and FFA levels. PLP enhances INS sensitivity by regulating key genes of hepatic glycolipid metabolism (e.g., inhibition of PEPCK, G6Pase, activation of GS, GLUT2, etc.) and by synergistically regulating the PI3K/Akt, FoxO1, Sterol Regulatory Element-Binding Protein-1 (SREBP-1)/ACC and PPAR-α/Low-Density Lipoprotein Receptor (LDLR) pathway to enhance INS sensitivity.

### 4.6. Improvement of Glucose Tolerance

Glucose tolerance reflects the body’s ability to regulate blood glucose levels after glucose intake. In patients with impaired glucose tolerance, the ability to store glucose as glycogen is reduced. Some NPs can improve glucose tolerance by activating signaling pathways such as PI3K/Akt, promoting GSK-3β phosphorylation, inhibiting GP activity, up-regulating GS expression, and increasing glycogen synthesis [108]. For example, *Gynura divaricata* (L.) DC polysaccharide significantly inhibited α-glucosidase activity and elevated glycogen content and PK and HK activities in IR-HepG2 cells. It regulates blood glucose by modulating the expression of genes related to PI3K/Akt, AMPK and GS/GSK-3β signaling pathways [109]. *Grifola frondosa* polysaccharide attenuated weight loss, hyperglycemia, and symptoms of abnormal glucose tolerance in T2D rats [110]. *Grifola frondosa* polysaccharide reduced hepatic pro-inflammatory factor levels, elevated the secretion of anti-inflammatory factors, inhibited hepatic macrophage infiltration, and improved IR. *Lycium barbarum* polysaccharides improved glucose tolerance in healthy mice by inhibiting the interferon-gamma (IFN-γ) signaling pathway, and also protected the INS content in the β-cell injury model [111].

NPs regulate glucose metabolism and maintain glucose homeostasis through a variety of pathways (Figure 2). NPs enhance insulin signaling by inhibiting JNK, releasing its inhibition of IRS, and activating pathways such as the PI3K/Akt pathway and AMPK. Insulin binds to the insulin receptor on the cell membrane and activates the IRS, which in turn activates the PI3K/Akt pathway. Akt inhibits GSK-3β to promote glycogen synthesis and inhibits FoxO1 to decrease the expression of enzymes related to gluconeogenesis (G6Pase, PEPCK). Akt also promotes the translocation of GLUT4 to increase glucose uptake and activates HK, GK, and promotes glycolysis to produce glucose 6-phosphate.

### 4.7. Protecting Pancreatic Beta Cells

In diabetes, pancreatic β-cell damage is mainly driven by oxidative stress, chronic inflammation, autophagy–apoptosis imbalance, and disturbances in gut microbiota. For example, excessive accumulation of ROS impairs mitochondrial function and activates apoptotic pathways. Defective cellular autophagy leads to accumulation of abnormal mitochondria and exacerbates apoptosis. Dysbiosis triggers a decrease in SCFAs and impairs GLP-1-mediated β-cell protection.

Some NPs protect β-cells through multi-targeted intervention mechanisms, e.g., inhibition of β-cell apoptosis, activation of antioxidant pathways, inhibition of inflammatory injury, modulation of cellular autophagy–apoptosis homeostasis, and regulation through the gut microbiota–islet axis [112]. Specifically, some NPs can protect pancreatic β-cells through the Nrf2/ARE antioxidant pathway, TLR4/MyD88/NF-κB inflammatory axis, AMPK/mTOR autophagy regulation pathway, and the microbiota–SCFAs–GLP-1 axis.

Some NPs reduce β-cell apoptosis by up-regulating the expression of anti-apoptotic protein Bcl-2 and down-regulating the level of pro-apoptotic protein Bax through the mitochondrial apoptotic pathway (Bcl-2/Bax–Caspase-3 axis), blocking the release of mitochondrial cytochrome C, and inhibiting the activation of caspase-3, which in turn reduces the apoptosis of β-cells. Bcl-2 blocks apoptosis by inhibiting the release of cytochrome C from mitochondria, and caspase-3 can cleave a variety of substrates to trigger cell disassembly, such as Poly (ADP-ribose) polymerase (PARP) [113]. Squash polysaccharide improved INS secretion in MIN6 cells in vitro. Squash polysaccharide increased INS secretion, decreased ATP, Ca^2+^, mitochondrial membrane potential and caspase-3 activity, and improved DNA damage, thus preventing apoptosis [114]. Zhao et al. [115] investigated the hypoglycemic mechanism of *Holothuria leucospilota* polysaccharide (HLP) in a spontaneous diabetic GK rat model. HLP significantly improved glucose tolerance abnormality, regulated lipid/hormone levels, and repaired pathological damage to the pancreas and colon (mucosal barrier). High-dose HLP (200 mg/kg) synergistically ameliorated dysglycemia by up-regulating PPAR-α/γ, PI3K/Akt/GLUT4 pathways, and anti-apoptotic factor Bcl-2, and down-regulating the expression of pro-apoptotic factor Bax and lipid transporter protein CD36. HLP also increased the abundance of SCFAs-producing bacteria in the feces, and reduced the number of opportunistic pathogens.

Some NPs may exert a dual protective effect on β-cells through antioxidant–anti-apoptotic synergism. Li et al. [116] investigated the molecular mechanism by which *Opuntia milpa alta* polysaccharide (MAP) protects β-cells from oxidative stress damage by activating the Nrf2 antioxidant pathway. It was shown that MAP significantly restored alloxan-induced INS-1 cell viability, reduced ROS, NO and MDA levels, and inhibited apoptosis by modulating the Bcl-2/Bax ratio and inhibiting the caspase cascade reaction. MAP enhances cellular oxidative defenses through the promotion of Nrf2 translocation in the nucleus and its downstream expression of antioxidant proteins, and simultaneously inhibits apoptosis execution molecules such as caspase-3.

Some NPs can promote autophagosomal clearance of dysfunctional mitochondria by activating hepatic AMPK and inhibiting mTOR signaling. Pectic bee pollen polysaccharide from *Rosa rugosa* can alleviate metabolic abnormality by regulating autophagy through the AMPK/mTOR pathway. It specifically enhances hepatic autophagy and promotes the expression of lipolytic enzymes and the clearance of dysfunctional mitochondria, thus improving hepatic steatosis, IR and glucose metabolism disorders in obese mice [117]. Ren et al. [118] investigated the repair mechanism of insulin secretion disorders by *Astragalus* polysaccharide (APS) through pancreatic β-cells induced by LPS. APS was able to reverse the inhibition of GLUT2, GK, PDX-1, and insulin gene expression by LPS and restore glucose-stimulated INS secretion. APS modulates GLUT2 expression through activation of the Akt/mTOR signaling axis, which in turn improves β-cell function.

### 4.8. Antioxidant Stress

Oxidative stress is a pathological state in which the rate of free radical (e.g., ROS) generation exceeds the scavenging capacity of the antioxidant defense system in response to endogenous stimuli in the body, leading to a dynamic imbalance between the oxidative and antioxidant systems, which is closely related to the development of T2D and its complications [119]. Excess free radicals can aggravate IR and tissue dysfunction by attacking biological macromolecules such as DNA, proteins, and lipids, triggering lipid peroxidation, protein inactivation, and mitochondrial damage cascade reactions. Some NPs may exert anti-oxidative stress effects by activating the endogenous antioxidant defense system, direct scavenging of free radicals, protection of mitochondrial function, inhibition of reduced nicotinamide adenine dinucleotide phosphate (NADPH) oxidase (NOX) activity, and inhibition of AGEs production.

Some NPs can up-regulate antioxidant enzymes such as SOD, glutathione peroxidase (GPx), and CAT to scavenge excess ROS by activating the Nrf2/ARE pathway and the Nrf2/Keap1 signaling pathway. Yang et al. [120] investigated the multidimensional mechanism of Pumpkin polysaccharides to ameliorate the disorders of glycolipid metabolism in diabetic mice. Pumpkin polysaccharides inhibit MDA accumulation by increasing SOD and CAT activities and GSH levels, significantly lowering blood glucose and enhancing glucose tolerance. *Moringa oleifera* leaf polysaccharide (279.48 kDa) possessed strong α-glucosidase inhibitory activity and antioxidant capacity. In high glucose-induced IR-HepG2 cells, polysaccharide intervention enhanced glucose consumption and glycogen synthesis, elevated CAT/SOD/GPx activity, and reduced MDA and ROS levels [121].

NOX is an enzyme that can catalyze the reduction of oxygen to generate the superoxide anion O^2−^ using NADPH as an electron donor. Excess fatty acids can indirectly promote the secretion of NOX by activating signaling pathways such as Protein Kinase C (PKC). Increased NOX further exacerbates the state of oxidative stress within the cell. Some NPs can reduce O^2−^ generation by down-regulating NOX activity [122]. Some NPs can restore PI3K/Akt signaling by inhibiting ROS-mediated serine phosphorylation of IRS-1. Zhu et al. [123] evaluated the anti-aging and hypoglycemic activities of *Enteromorpha prolifera* polysaccharide (EPP) by an aged diabetic mouse model. EPP can inhibit the expression of aging-related genes (p38 MAPK, NADPH oxidase 1 (NOX-1)), reduce oxidative stress and inflammation, and protect liver function. EPP also improves glucose metabolism disorders and reshapes the intestinal micro-ecological balance through enrichment of Lactobacillus and other beneficial bacteria. Wu et al. [124] investigated the hypoglycemic mechanism of azuki bean polysaccharide (AB-P) in HFD combined with STZ-induced T2D rats. AB-P significantly reduced FBG and lipids, enhanced hepatic glycogen storage and improved INS sensitivity, and inhibited apoptosis of pancreatic β-cells. AB-P promotes glucose metabolism regulation through activation of the PI3K/Akt/GLUT2 signaling pathway, and enhances antioxidant capacity to mitigate diabetes-associated oxidative damage. The results showed that AB-P can significantly reduce glucose metabolism disorders, enhance hepatic glycogen storage and improve INS sensitivity, and inhibit pancreatic β-cell apoptosis.

AGEs are cross-linked compounds generated by non-enzymatic reactions between glucose or its degradation products and free amino groups of biological macromolecules, such as proteins and nucleic acids, in a hyperglycemic environment. AGEs activate the NF-κB signaling pathway by binding to Receptor for Advanced Glycation End-products (RAGE), and promote the expression of inflammatory factors while inducing oxidative stress. Some NPs can down-regulate the expression of inflammatory factors induced by AGEs through the blockade of AGEs formation and inhibition of the RAGE–NF–κB signaling axis [125]. *Lentinus edodes* mycelium polysaccharide (LMP) alleviated AGEs-induced pyroptosis injury in human umbilical vein endothelial cells. LMP down-regulated the pro-pyroptosis factor Metastasis through inhibition of the NLRP3/Caspase-1/gasdermin D (GSDMD) pyroptosis pathway and down-regulated the expression of pro-pyroptosis factors Metastasis-associated lung adenocarcinoma transcript 1 (MALAT1) with mTOR expression, and up-regulated microRNA-199b (miR-199b) levels, ameliorating diabetic vasculopathy [126].

## 5. Regulation of Immunity

The pathological process of diabetes mellitus is closely associated with dysregulation of immune homeostasis, and this association cuts across the full-cycle regulatory network of IR, β-cell dysfunction, and metabolic disorders. Chronic high-sugar and high-fat diets produce large amounts of ROS that exceed the body’s ability to clear them, e.g., excess fatty acids lead to an increase in ROS. After entering the mitochondria, excess fatty acids are oxidized in the mitochondria to generate acyl coenzyme A (Acyl CoA), which in turn generates acetyl-coenzyme A (Acetyl-CoA) through the process of β-oxidation, and these substances participate in cellular energy metabolism as the substrates of the tricarboxylic acid cycle (TCA cycle). However, during this process, the mitochondrial electron transport chain is perturbed, which can lead to electron leakage and ROS production.

Accumulation of ROS disrupts the mitochondrial membrane potential of pancreatic β-cells and inhibits ATP synthesis, which in turn inhibits the synthesis and secretion of INS [127]. ROS also activate inflammatory signaling pathways such as NF-κB, which induces the release of inflammatory factors, such as TNF-α and IL-6.

Under inflammation, pro-inflammatory factors such as TNF-α and IL-6 can exacerbate oxidative stress, causing the body to produce excessive ROS. Inflammation-related ROS accumulation also disrupts the mitochondrial membrane potential and exacerbates oxidative stress, which in turn activates the inflammatory pathway, forming a vicious cycle that exacerbates IR and glucose metabolism abnormalities. TNF-α activates the JNK pathway, resulting in the phosphorylation of IRS-1 serine (Ser307) and impeding PI3K/Akt signaling. Various pro-inflammatory factors also damage pancreatic β-cells and mitochondrial function, interfering with insulin signaling and leading to decreased INS secretion [128].

As a core pathological driver of IR and pancreatic β-cell dysfunction, chronic low-grade inflammation plays a key role in the development of T2D. Inhibition of the inflammatory response is a key component in the hypoglycemic mechanism of NPs. NPs can exert hypoglycemic effects by inhibiting the inflammatory response in multiple dimensions, including modulation of the gut microbiota–immune axis, remodeling of the immune microenvironment, modulation of the signaling pathway, and inhibition of inflammatory vesicles. Some NPs reduce NO and prostaglandin E2 (PGE2) production by remodeling the immune microenvironment, inhibiting the TLR2/TLR4/MyD88 pathway in macrophages/dendritic cells, and down-regulating inducible nitric oxide synthase (iNOS) and cyclooxygenase-2 (COX-2). Some NPs activate the Nrf2 pathway and enhance endogenous antioxidant defense. Nrf2 inhibits NF-κB-mediated inflammatory response, reduces the levels of inflammatory factors such as IL-6 and TNF-α, and blocks the vicious cycle of ROS and inflammation.

Some NPs can inhibit K^+^ efflux or improve mitochondrial function by regulating ion channels to reduce ROS generation, prevent oligomerization of NLRP3 with apoptosis-associated speck-like protein containing a CARD (ASC) and caspase-1 activation, and thus inhibit interleukin-1β (IL-1β) and interleukin-18 (IL-18) maturation and release. Wang et al. [129] investigated the ameliorative effects and mechanisms of *Ganoderma atrum* polysaccharide (PSG) on diabetic cardiomyopathy in T2D rats. PSG activated the Nrf2 pathway and attenuated myocardial oxidative damage. PSG inhibited NLRP3 inflammasome activation and TLR4/NF-κB signaling, blocking the inflammatory cascade response. PSG remodeled the metabolic profile of gut microbiota, increased SCFAs, decreased LPS, and mediated cardioprotective effects through metabolite–inflammatory interactions.

Some NPs also inhibit the activation of JNK/p38/ERK, a key node of the MAPK pathway, and down-regulate the phosphorylation level of STAT3 in JAK-STAT signaling, resulting in multi-targeted inhibition of inflammation-related gene transcription. Some NPs restored PI3K/Akt signaling by up-regulating mitogen-activated protein kinase phosphatase-1 (MKP-1), dephosphorylating JNK, and blocking its mediated phosphorylation of the serine 307 site of IRS-1. Ruan et al. [130] investigated the mechanism of action of *Ulva lactuca* polysaccharide (ULP) to ameliorate symptoms in aged diabetic mice. ULP improved glucose metabolism by activating the INSR/AMPK pathway to enhance INS sensitivity and inhibiting the JNK/JAK/STAT3 inflammatory senescence pathway. ULP remodeled the gut microecology by enriching *Alloprevotella* (SCFAs-producing) and *Pediococcus* (probiotic), which indirectly enhanced the antioxidant and metabolic homeostasis of the host. Liu et al. [131] extracted a heteropolysaccharide, APS-1I (17.0 kDa), and a linear dextran, APS-2II (10.0 kDa), from *Angelica sinensis*, and found that APS-1I has a stronger affinity for RAGE. APS-1I has a stronger affinity for p-IRS-1 (Ser307), p-IRS-2 (Ser731), phosphorylated JNK (p-JNK), and phosphorylated p38 MAPK (p-P38). APS-1I significantly promoted glucose uptake in IR cells by inhibiting RAGE-mediated phosphorylation of the JNK/p38-IRS signaling pathway.

Some NPs protect mitochondrial structure and function and maintain membrane potential to reduce inflammatory signals induced by oxidative damage. Some NPs may also reduce lipid peroxidation by decreasing MDA content in vivo. Zhong et al. [132] isolated four polysaccharides from *Flos sophorae immaturus* and found that they improved IR. On the one hand, these four polysaccharides promote glucose catabolism by activating the AMPK pathway to enhance glycolysis and inhibit gluconeogenesis. On the other hand, the four polysaccharides up-regulated glycogen synthesis through the IRS-1/PI3K/Akt pathway and enhanced anabolism. Meanwhile, they reduced ROS and MDA levels and elevated SOD and CAT activities, alleviating oxidative damage. Among them, galacturonic acid-rich polysaccharides exhibited stronger anti-IR activity due to their structural specificity.

NPs can maintain the stability of the intracellular environment through multiple pathways in response to oxidation, inflammation and apoptosis triggered by hyperglycemia and excess fatty acids (Figure 3). AGEs produced by hyperglycemia act via RAGE, and excess fatty acids affect Acyl CoA activity, both of which lead to ROS production. In addition, excess fatty acids produce O^2−^ via NOX, which further affects mitochondria and related apoptotic proteins. Some NPs activate the Nrf2/ARE pathway and promote the expression of antioxidant enzymes such as SOD, GPx, and CAT to counteract oxidative stress. Some NPs inhibit the NF-κB pathway and reduce the release of inflammatory factors such as TNF-α and IL-6 to reduce inflammation. Some NPs can act on P38, Bax, Bcl2, caspase-3, etc., to regulate apoptosis.

## 6. Clinical Application Strategies

### 6.1. Delivery System

Natural polysaccharides, with their excellent biocompatibility, degradability and functional diversity, have been developed as the core materials for clinical drug delivery systems, which can be adapted to a variety of drug delivery routes, such as oral, injectable, topical and inhalation. Natural polysaccharide-based delivery systems come in various forms, such as nanoparticles, microspheres, hydrogels, liposomes, functionalized membrane materials, and prodrug couplers.

Currently, various NPs nano-delivery systems, such as nanoliposomes, nanoparticles, nanomicelles, nanoemulsions and nanohydrogels, are commonly used in clinical practice. These delivery systems have demonstrated significant advantages in the fields of oncology, metabolic diseases and tissue repair by modulating drug release kinetics, enhancing targeting and improving bioavailability. Chuang et al. [133] constructed a self-directed nanocarrier based on fucoidan/glycol chitosan. This carrier integrates the targeting of polysaccharides, significantly enhances the accumulation of drugs in the target area, promotes tissue repair and inhibits inflammation, and can be used for the treatment of diabetic nephropathy. Fan et al. [134] constructed a nanodelivery system by combining Mn^2+^ with tea polysaccharides, and significantly improved the IR of T2D mice through the synergistic regulation of glucose and lipid metabolism by targeting delivery of Mn^2+^ and tea polysaccharides. Mn^2+^ plays the dual roles of cross-linking agent and functional synergist: on the one hand, it forms stable nanostructures by binding to the protein and glucuronide groups of tea polysaccharides, and on the other hand, it can activate the CaMKII pathway to promote INS secretion.

By preparing pH-responsive nanocarriers, NPs can be protected from enzymatic degradation in the upper digestive tract to ensure that they are released to play a role at specific sites. Guo et al. [135] designed a bifunctional hybrid micellar system based on *Angelica* polysaccharide and *Astragalus* polysaccharide. This system achieved precise enrichment of *Angelica* polysaccharide at inflammatory foci of diabetic nephropathy by integrating pH-responsive release with salivary acid-mediated E-selectin receptor-targeted delivery.

Chitosan hydrogels can be used for drug delivery, and their good biocompatibility and antimicrobial activity can protect tissues while achieving slow drug release and prolonging the duration of action. For example, Li et al. [136] constructed a multifunctional hydrogel by cross-linking *Bletilla striata* polysaccharide, carbomer 940 and carboxymethyl chitosan. The hydrogel accelerated wound healing in diabetic mice through multiple mechanisms such as synergistically promoting neovascularization, modulating the inflammatory microenvironment, and inducing collagen regeneration, providing a novel dressing for diabetic wound management. Tang et al. [137] developed a multinetwork hydrogel based on *Astragalus* polysaccharide, chitosan, and alginate. The hydrogel realized the dual functions of diabetic foot ulcer repair and electromyographic signal monitoring by embedding conductive nanoparticles loaded with resveratrol. The hydrogel combined the hypoglycemic and anti-inflammatory properties of *Astragalus* polysaccharide, the antioxidant pro-angiogenic effect of resveratrol, and the mechanical enhancement and conductive properties of the conductive nanoparticles to synergistically promote wound healing.

However, the production scale-up, in vivo long-term toxicity assessment and stability optimization of NPs delivery systems for clinical applications still require in-depth studies. In this regard, the stability and targeting of NPs can be enhanced by chemical modification (e.g., sulfation, acetylation) or compounding other materials (e.g., synthetic polymers).

### 6.2. Design of Combination Drug Programs

Synergistic effects of some NPs with synthetic drugs such as metformin can lower drug doses and reduce side effects. For example, NPs works synergistically to exert hypoglycemic effects by down-regulating hepatic gluconeogenic enzymes, while metformin inhibits hepatic glucose output. Wu et al. [138] found that *Sargassum fusiforme* polysaccharide can be used as an adjuvant to metformin for hypoglycemia. Experiments showed that the combination of the two substances exerted a better hypoglycemic effect compared with metformin. The main reason is that *Sargassum fusiforme* polysaccharide down-regulated the expression of 3-hydroxy-3-methylglutaryl coenzyme A (HMG-CoA), PEPCK and G6Pase, and up-regulated the expression of hepatic CYP7A1. Sun et al. [139] investigated the effect of a *Naeematelia aurantialba* polysaccharide (NAP) co-administration program. In vitro experiments showed that NAP had comparable inhibitory activity to acarbose on α-glucosidase and α-amylase. In diabetic mouse models and IR-HepG2 cells, the combination of NAP and metformin demonstrated synergistic hypoglycemic effects, significantly lowering body weight, serum INS levels, improving glucose/insulin tolerance and enhancing antioxidant capacity compared to the single-agent group. Long et al. [140] found that the addition of *Astragalus* polysaccharide to low-dose metformin may be an effective strategy in the treatment of T2D.

Diabetic patients are often associated with dyslipidemia, and the combination of NPs with lipid-lowering drugs can improve both blood glucose and lipid metabolism. The combination of NPs with probiotics/synbiotics can also synergistically improve glucose metabolism. For example, pomelo peel polysaccharides increased the expression of tight junction proteins in the gut tract, reduced the release of inflammatory factors, and promoted the colonization of probiotics, such as *Lactobacillus acidophilus*, in mice [141]. Mushroom polysaccharides and probiotics can be used as pharmaceutical excipients in the development of a nanodelivery system [142].

However, before combining drugs, research organizations must conduct clinical trials of NPs in combination with drugs such as metformin to monitor patients’ glycemic changes. For example, pharmacokinetic studies of metformin–polragalus polysaccharide combinations must be designed to focus on assessing the dose effect of reduced hepatic and renal toxicity after administration of the combination.

### 6.3. Individualized Treatment Strategies

Individual differences in the gut microbiota of diabetic patients can lead to uneven NPs efficacy, and precise intervention strategies based on microbiota typing may become an important research direction in the future. In clinical treatment, by analyzing key microbiome characteristics such as the *Firmicutes*/*Bacteroidetes* ratio and the abundance of SCFAs-producing genera through 16S rRNA sequencing, patients can be classified into different enterotypes, thereby enabling targeted selection of polysaccharide types for treatment. For example, highly branched structural polysaccharides may have a superior modulating effect on patients with lower microbiota diversity. Clinically, we can also carry out large-scale cohort studies to establish an NPs efficacy prediction model based on the diversity of intestinal microbiota, host BMI and other factors, which can intelligently recommend appropriate polysaccharides for patients with specific microbiota characteristics, so as to achieve the individualized therapeutic goal of precise drug delivery.

## 7. Discussion

Current research on the hypoglycemic mechanism of NPs lacks in-depth investigation of its complex structure and quantitative correlation of synergistic effects across organs and pathways, resulting in a lack of systematic theoretical guidance for structural optimization. In this regard, researchers can employ multi-omics techniques to elucidate the structure–activity relationships of NPs. For example, by spatial metabolomics techniques based on mass spectrometry imaging (MSI), the distribution characteristics of NPs in the gut–liver–islet axis are localized. Screening for NPs-regulated β-cell-specific signaling pathways can be accomplished by utilizing single-cell transcriptomics. By combining metabolomics (e.g., LC-MS/MS) and macrogenomics (16S rRNA sequencing), it is possible to analyze the differences in the interventions of different structural NPs on metabolites (SCFAs, BAs) and signaling pathways in the gut microbiota [143]. By utilizing molecular docking and molecular dynamics simulation techniques, the binding patterns of NPs to key targets are predicted to guide branched modifications or chemical group introduction [144]. For example, Yang et al. [145] investigated the conformational relationship of Dendrobium acetylated mannans (DDAMs) and their regulatory mechanism on insulin stability. The experiments revealed the law of directional regulation of the distribution of acetylation sites by glycosidic bond type through the enzymatic–molecular dynamics coupling technique. It was found that polysaccharides with a high degree of acetylation significantly enhanced the thermal stability of insulin conformation and promoted hypoglycemic activity, and their efficacy was positively correlated with acetylation level.

The hypoglycemic effect of some NPs varies widely between in vitro cellular models and animal models. Route of administration may also limit the study of the hypoglycemic effects of NPs. The bioavailability of oral polysaccharides is affected by the metabolism of gut microbiota, while most in vitro studies add polysaccharides directly to the cell culture medium, ignoring the first-pass effect. Three-dimensional organoid models (e.g., gut organoids, pancreatic islet organoids) can mimic the effects of NPs on complex tissues and make up for the limitations of the traditional cellular models [146]. For example, using gut organoids to study the regulatory mechanism of NPs on tight junction proteins and mucus layer can provide a basis for the development of new gut protective agents. Ding et al. [147] found that *Astragalus* polysaccharide can be used to protect against gut injuries induced by ionizing radiation by promoting the regeneration of gut stem cells through organoid experiments. Wang et al. [148] systematically revealed the stability and bioactivity-enhancing effects of *Levilactobacillus brevis* M-14 exopolysaccharide during digestion by modeling static digestion in vitro. Song et al. [149] demonstrated the low intestinal absorption of *Astragalus membranaceus* polysaccharides using a Caco-2 monolayer cell model, and found the relevant intestinal metabolic pathways through an in vitro digestion–fermentation model.

In addition, most animal experiments used the STZ-induced T1D model, but the direct damage of STZ on pancreatic islet β-cells may mask the indirect protective effect of NPs on the gut microbiota, whereas the HFD-induced T2D model is more closely related to the clinical IR characteristics but suffers from the problems of long modeling period and large individual differences. Although the cross-organ effects of NPs via the gut–hepatic and microbiota–immune axes have been demonstrated, the effect of the spatiotemporal dependence of signaling between different organs has not been clarified; for example, how SCFAs regulate the spatiotemporal dynamics of hepatic gluconeogenesis via the gut–hepatic axis has not been elucidated. These issues need to be discussed in depth in future studies.

## 8. Conclusions

Natural polysaccharides (NPs), as a class of natural bioactive molecules with multitarget regulatory potential, exhibit unique hypoglycemic potential through their distinctive structure–activity relationships, cross-organ interaction networks, and synergistic antioxidant–inflammatory mechanisms. Molecular weight, branch complexity, and chemical modifications determine the action pathways by which NPs exert hypoglycemic activity by regulating solubility, target specificity, and gut microbiota metabolic characteristics. Low molecular weight NPs can directly target pathways such as α-glucosidase, PI3K/Akt, and AMPK to rapidly inhibit glucose absorption and promote glycogen synthesis. In contrast, high molecular weight NPs rely more on gut microbiota fermentation to generate SCFAs, enabling long-acting regulation of GLP-1 secretion and insulin sensitivity. Branch complexity and chemical modifications further enhance the hypoglycemic activity of NPs by optimizing target-binding ability and bioavailability.

In terms of metabolic and immune regulation, NPs promote glucose uptake by activating insulin signaling pathways (INSR/IRS/PI3K/Akt), inhibit FoxO1/G6Pase-mediated gluconeogenesis, and protect pancreatic β-cell function through dual mechanisms. The gut microbiota acts as a core hub, not only mediating the conversion of NPs to SCFAs but also collaboratively improving insulin resistance (IR) and chronic inflammation by repairing the intestinal barrier and inhibiting LPS translocation. This forms a “microbiota–metabolism–immunity” regulatory network. Additionally, NPs scavenge ROS via the Nrf2/ARE pathway, inhibit the AGEs-RAGE-NF-κB inflammatory cascade, and block the vicious cycle of “oxidation–inflammation–metabolic disorder”.

Despite the gradual clarification of the multidimensional hypoglycemic mechanisms of NPs, their clinical translation still faces challenges such as precise resolution of structure–activity relationships, dynamic modeling of cross-organ interactions, and optimization of bioavailability. Future research should integrate multi-omics technologies (e.g., metagenomics, metabolomics) and organoid models to decipher the spatiotemporal action patterns of NPs along the gut–liver–islet axis. By combining molecular docking and targeted modification technologies for structural optimization, efforts can be made to advance the development of personalized intervention strategies based on microbiota typing. The multitarget synergistic properties of NPs offer new directions for diabetes treatment, holding promise as safe and efficient novel candidate drugs.

## Figures and Tables

**Figure 1 molecules-30-01980-f001:**
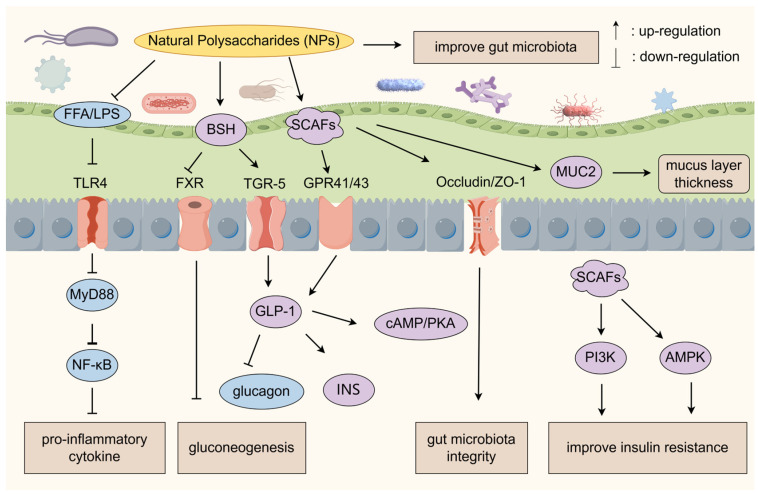
Some mechanisms of NPs regulating gut microbiota and metabolic–inflammatory pathways.

**Figure 2 molecules-30-01980-f002:**
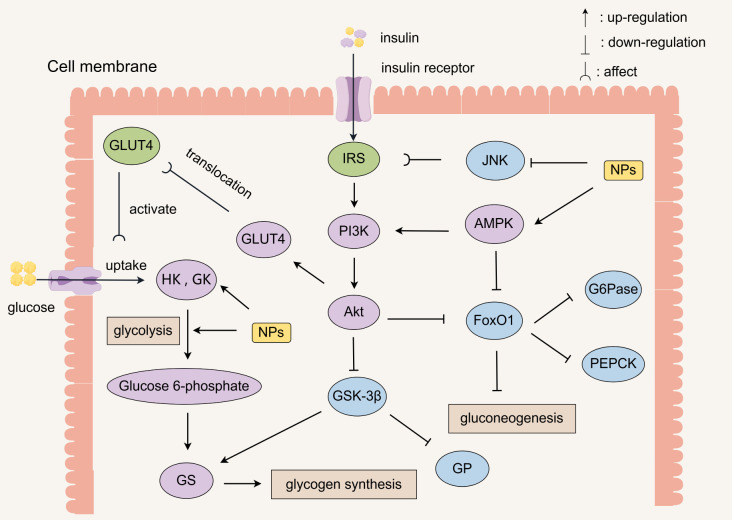
Some mechanisms of NPs regulation of insulin signaling pathway and glucose metabolism.

**Figure 3 molecules-30-01980-f003:**
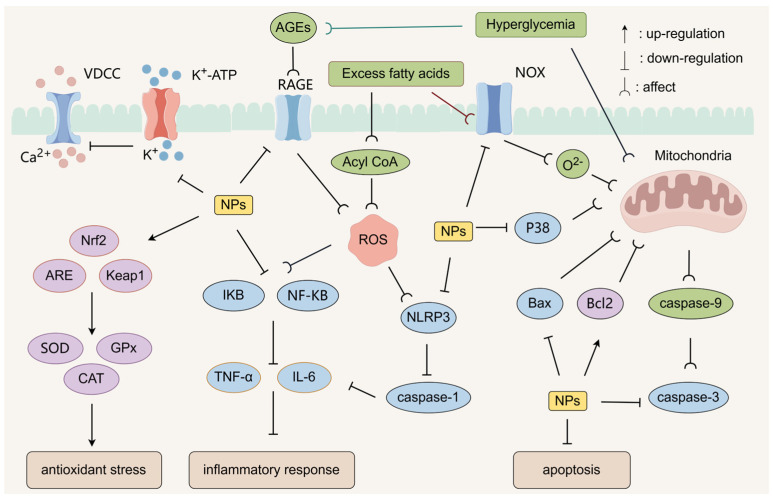
Some mechanisms by which NPs regulate oxidative stress, inflammation, and apoptosis related to diabetes.

**Table 1 molecules-30-01980-t001:** Effect of molecular weight size on the hypoglycemic activity of polysaccharides.

Polysaccharide Type	Molecular Weight (kDa)	Experimental Model	Core Mechanisms	Reference
*Fructus corni* polysaccharides	59	IR-HepG2 cells, T2D mice	GLUT2 ↑, GK ↑	[22]
Pumpkin polysaccharides	33.72	Acute diabetic mice	SOD ↑, CAT ↑, GSH ↑, PK ↑, PEPCK ↓	[23]
*Euglena gracilis* polysaccharides	130.8	IR-HeoG2 cells	PI3K ↑, Akt ↑, GLUT4 ↑	[24]
*Abelmoschus esculentus* L. Moench (okra) polysaccharide	3.02 × 10^3^	T2D mice	GSK-3β ↑, GS ↑, PI3K ↑, Akt ↑	[25]
*Panax notoginseng* polysaccharide	8.27	High-fat diet mice	GLUT2 ↓, SGLT-1 ↓. p-IRS ↑, p-AMPK ↑	[26]

Table notes: GK (Glucokinase), SOD (Superoxide Dismutase), CAT (Catalase), GSH (Glutathione), PK (Pyruvate Kinase), GSK-3β (Glycogen Synthase Kinase 3 Beta), SGLT-1 (Sodium-Glucose Cotransporter 1), p-IRS (Phosphorylated Insulin Receptor Substrate), p-AMPK (Phosphorylated AMP-activated protein kinase), ↑: up-regulated, ↓: down-regulated.

**Table 2 molecules-30-01980-t002:** Regulatory effects and mechanisms of NPs on key enzymes in blood glucose metabolism.

Enzyme	Effect	Mechanism	Direction of NPs Regulation
GK	Catalyzes the phosphorylation of glucose to G6P	INS activates GK through the PI3K/Akt pathway	Activate
G6Pase	Catalyzes the hydrolysis of G6P to glucose	FoxO1 transcriptionally activates G6Pase	Inhibit
GS	Promotes glycogen synthesis	Inhibition of GSK-3β by INS activation of the PI3K/Akt pathway deregulates the inhibition of GS phosphorylation	Activate
GP	Catalyzes the breakdown of glycogen to glucose-1-phosphate	Activation of PKA pathway by glucagon	Inhibit
HK	Catalyzes the phosphorylation of glucose to G6P	Insulin signaling pathway	Activate
PK	Catalyzes the formation of pyruvate from phosphoenolpyruvate (PEP)	Activated by FBP-1 (fructose-1,6-bisphosphate) denaturation	Activate
GSK-3β	Phosphorylation inhibits GS and blocks glycogen synthesis	INS phosphorylates GSK-3β through the PI3K/Akt pathway and inactivates it	Inhibit

## Data Availability

Not applicable.

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
