# Peer review of "Mechanistic Advances in Hypoglycemic Effects of Natural Polysaccharides: Multi-Target Regulation of Glycometabolism and Gut Microbiota Crosstalk"

_molecules, 2025, doi:10.3390/molecules30091980_

Round 1

Reviewer 1 Report

Comments and Suggestions for Authors

The submitted review manuscript “Mechanistic Advances in Hypoglycemic Effects of Natural Polysaccharides: Multi-Target Regulation of Glycometabolism and Gut Microbiota Crosstalk” summarizes investigations on the topic over the past five years. More than one hundred authors are cited. The selection of the cited articles is appropriate and thematically relevant. The manuscript is generally well-written and well-structured, with well-selected and appropriately titled subsections. The topic is actual and interesting. The application of natural polysaccharides continues to be the subject of numerous current research because they are a promising strategy for combating various metabolic disorders, including insulin resistance and diabetes. However, it should be noted that several similar review articles have been published in recent years, some of which are considerably more comprehensive and understandable. Therefore, the manuscript's present form does not substantially advance the current understanding or offer a significant synthesis of existing knowledge. It needs to be significantly improved to fulfill the standards of a high-quality review and intrigue the reader. In my opinion, a significant revision is recommended, and several essential weaknesses need to be addressed:

Major revisions:

Aim: The aim of the article is clearly formulated. However, within the main text, it remains unclear what specific recommendations are provided regarding "structural optimization" and "clinical application." In fact, in conclusion, the authors acknowledge that "clinical application still faces many challenges such as the precise analysis of the complex structure-activity relationship," which further emphasizes the absence of concrete proposals or strategies in these key areas.

Main text: I find the main weakness of the presented review to be the fact that most of the subsections are focused on the description of well-known mechanisms and metabolic pathways. At the same time, the effects of natural polysaccharides are briefly mentioned, primarily through listing scientific facts without critical or comparative analysis. Key methodological details, such as the administered dose, route of administration, details of the experimental model, and the relationship between the molecular structure of the supplements and their biological effects, are missing. The entire text should be rewritten to provide methodologies used in the cited studies and a more detailed comparative, critical analysis of their results. This would significantly strengthen the scientific value and originality of the review.

Figures and tables: The figures presented are difficult to understand and lack clarity. They appear to visualize natural polysaccharides "in general," which is inappropriate in this context. Moreover, figures are referenced too early in the text relative to their actual placement (e.g., Figure 2 is mentioned on page 6 but only appears on page 12), which disrupts the flow of reading. Schematic representations illustrating the potential metabolic pathways affected by specific polysaccharides organized by molecular weight and/or chemical structure would be more valuable for the readers.

The single table provided an interesting overview of key hepatic enzymes but lacked specific information regarding the influence of natural polysaccharides on these enzymes. Comparative tables summarizing the natural supplements used in the referenced studies, including specific information such as molecular weight, chemical structure, and observed effects, would greatly enhance the clarity and utility of the review.

The conclusion does not adequately address, nor does it speculate on, the key objectives outlined earlier in the manuscript. It also fails to clearly identify the existing gaps in knowledge or propose future directions for experimental research in the field. It is recommended to be substantially revised.

Minor revisions:

  1. Lines 40–41: As currently stated, the sentence appears inaccurate. Please clarify or revise the statement.
  2. Please check reference 17, as the information attributed to it seems to not appear in the cited text.
  3. Please correct the formatting throughout the manuscript. There are inconsistencies, such as missing spaces before in-text citations and unnecessary spaces after them.
  4. Рevise the text carefully, as some sentences incorrectly begin with a lowercase letter instead of capital ones.
  5. Please check the use of definite and indefinite articles ("the," "a," "an").
  6. Please ensure that all abbreviations are clearly defined at their first occurrence in the text and used consistently throughout the manuscript.

Author Response

Response to Reviewer 1 Comments

Dear Reviewers,

Thank you for your positive comments and constructive feedback on our manuscript. We highly value your insights and have carefully addressed all the points raised. The revisions we made are highlighted in red throughout the manuscript, and we have also added new sections and content where necessary. Below, we provide a detailed Response  to each of your comments.

Major revisions:  

Comments 1: Aim: The aim of the article is clearly formulated. However, within the main text, it remains unclear what specific recommendations are provided regarding "structural optimization" and "clinical application." In fact, in conclusion, the authors acknowledge that "clinical application still faces many challenges such as the precise analysis of the complex structure-activity relationship," which further emphasizes the absence of concrete proposals or strategies in these key areas.

Response 1: Thanks to the insightful comments on the structural optimization and clinical application sections. We've made the following revisions: We've reworked and expanded the chapters on structural optimization and clinical application, adding specific recommendations. In Chapter 2, we now offer an in-depth discussion of the standardization of molecular weight, branching structure, and chemical modification, supported by additional references. A critical analysis of these factors has also been included at the end of each respective section. The relevant additions can be found on page 5, lines 205 - 208; page 6, lines 244 - 248; and pages 7 - 8, lines 327 - 333. To clarify the relationship between the molecular weight of polysaccharides and their hypoglycemic effects, we've included new content on pages 190 - 198. We've also added a new chapter on "6. Clinical Application Strategies", which consists of three sub - sections: "6.1 Delivery System", "6.2 Design of Combination Drug Programs", and "6.3 Individualized Treatment Strategies". Each sub - section comes with concrete recommendations. These can be found on page 25, lines 1105 - 1109; page 26, lines 1139 - 1143; and page 26, lines 1153 - 1157 respectively. All modifications are highlighted in red in the manuscript for easy identification.

Comments 2: Main text: I find the main weakness of the presented review to be the fact that most of the subsections are focused on the description of well-known mechanisms and metabolic pathways. At the same time, the effects of natural polysaccharides are briefly mentioned, primarily through listing scientific facts without critical or comparative analysis. Key methodological details, such as the administered dose, route of administration, details of the experimental model, and the relationship between the molecular structure of the supplements and their biological effects, are missing. The entire text should be rewritten to provide methodologies used in the cited studies and a more detailed comparative, critical analysis of their results. This would significantly strengthen the scientific value and originality of the review.

Response 2:We acknowledge the importance of providing a comprehensive understanding of hypoglycemic mechanisms of natural polysaccharides while avoiding mere compilation of existing knowledge. Our review manuscript aims to offer valuable theoretical support and guidance for future research directions in the structural optimization and clinical translation of natural polysaccharides for hypoglycemic applications.

To achieve this, we've adopted an innovative approach by organizing the content around the "structure - microbiota - metabolism - immunity - clinical" framework. This approach allows for a more integrated and in-depth exploration of the subject matter. We've made sure to avoid simplistic listings of research findings by instead emphasizing the cross-organ connections and interactions that underlie the hypoglycemic effects of natural polysaccharides.

In the revision process, we've rephrased the entire text of the review to present the information in a fresher and more coherent manner. Additionally, we've endeavored to incorporate more methodological details throughout the manuscript. This was done to provide a more comprehensive understanding of the research methods and approaches used in the field, which can aid readers in better grasping the intricacies of the studies discussed.

Comments 3: Figures and tables: The figures presented are difficult to understand and lack clarity. They appear to visualize natural polysaccharides "in general," which is inappropriate in this context. Moreover, figures are referenced too early in the text relative to their actual placement (e.g., Figure 2 is mentioned on page 6 but only appears on page 12), which disrupts the flow of reading. Schematic representations illustrating the potential metabolic pathways affected by specific polysaccharides organized by molecular weight and/or chemical structure would be more valuable for the readers.

Response 3:We have reprocessed the content of the three figures by removing redundant information and simplifying annotations to focus solely on the regulatory direction of natural polysaccharides (NP), enhancing clarity and accessibility. In the revised figures, complex regulatory relationships between targets are replaced with generalized "influence" indicators, accompanied by descriptive captions added above each figure to improve interpretability. For instance, as illustrated in Figure 1, "NP strengthens intestinal barrier integrity to limit harmful substance infiltration, suppresses the TLR4/MyD88/NF-κB pathway to attenuate inflammatory cytokine release, and promotes SCFAs production via probiotic metabolism. These SCFAs regulate glucose homeostasis and insulin sensitivity, while NP further modulates lipid metabolism and gluconeogenesis via BSH/FXR signaling, synergistically achieving hypoglycemic effects."

The updated figures are now strategically positioned immediately below their corresponding descriptive text in the manuscript (Page 12, Line 550; Page 19, Line 848; Page 24, Line 1056), ensuring logical flow and reader-friendly navigation. Revised figure titles explicitly highlight NP's mechanistic roles, aligning with the streamlined visual representations.

Comments 4: The single table provided an interesting overview of key hepatic enzymes but lacked specific information regarding the influence of natural polysaccharides on these enzymes. Comparative tables summarizing the natural supplements used in the referenced studies, including specific information such as molecular weight, chemical structure, and observed effects, would greatly enhance the clarity and utility of the review.

Response 4:We’ve added Table 1 in the chapter on molecular weight to illustrate the hypoglycemic effects of polysaccharides with different molecular weights. This addition aims to clarify the relationship between molecular weight and hypoglycemic mechanisms. Due to the limited literature providing key structural details like molecular weight and monosaccharide composition, we couldn’t fully tabulate the structure-activity relationship of NPs. Most hypoglycemic studies don’t specify this structural information. So, our current analysis focuses on the impact of molecular weight on hypoglycemic mechanisms, and we plan to expand this section as more detailed structural data becomes available.

Comments 5: The conclusion does not adequately address, nor does it speculate on, the key objectives outlined earlier in the manuscript. It also fails to clearly identify the existing gaps in knowledge or propose future directions for experimental research in the field. It is recommended to be substantially revised.

Response 5:To address knowledge gaps and suggest future research directions, we've added Chapter 7: "Discussion." This chapter offers specific strategies for structural optimization and multi-organ blood sugar regulation, and includes a critical analysis of the manuscript's content. It explains how our study systematically explores NP's structure-activity relationships, gut microbiota modulation, and signal pathway activation, all centered on the core goal of "multi-target coordinated blood sugar regulation," thus aligning with the research objectives outlined in the introduction.

In the discussion, we highlight three key limitations:

- The structure-activity relationship analysis is currently based on single parameters and doesn't connect complex structures with multi-pathway coordination.

- The dynamic integration of the cross-organ regulatory network (e.g.,gut - liver axis signaling thresholds) remains unclear.

- The molecular basis for probiotics-polysaccharide matching in personalized treatment is ill-defined.

We also propose future research directions, emphasizing the combination of technological innovation and clinical needs. For example, we suggest using pH-responsive carriers to improve polysaccharide delivery efficiency in the gut, and conducting combination drug trials to reduce existing drug side effects. These suggestions enhance the translational value of the research and ensure our proposals are practical and actionable.

Minor revisions:  

Comments 6: Lines 40–41: As currently stated, the sentence appears inaccurate. Please clarify or revise the statement.

Response 6:We've revised the unclear statement and expanded the text (lines 78 - 97) to detail how NPs exert hypoglycemic effects in the body. To strengthen this section, we've also incorporated new references.

Comments 7: Please check reference 17, as the information attributed to it seems to not appear in the cited text.

Response 7:We’ve enhanced the text (lines 231 and 236) by incorporating a more relevant study and expanding the discussion. The revised text now states: “Dendritic cell surface C-type lectin receptors (e.g., Dectin-1) recognize highly branched NPs, triggering endocytosis and activating M2 polarization signals. The glucose residues at the branch ends of highly branched β-(1→3)/(1→6)-glucans (e.g., Ganoderma lucidum polysaccharides) specifically bind to the macrophage Dectin-1 receptor. This interaction activates the PI3K/Akt pathway and enhances insulin signaling.”

Comments 8: Please correct the formatting throughout the manuscript. There are inconsistencies, such as missing spaces before in-text citations and unnecessary spaces after them.

Response 8:We have corrected space formatting errors throughout the text citation and have done a detailed inspection.

Comments 9: Ð evise the text carefully, as some sentences incorrectly begin with a lowercase letter instead of capital ones.

Response 9:We have corrected this part by setting the format of capitalizing the beginning of the first letter of a sentence.

Comments 10: Please check the use of definite and indefinite articles ("the," "a," "an").

Response 10:We have corrected definite and indefinite articles that were unclear. For example, we found more phrases for the polysaccharide, and we now use the abbreviation for the polysaccharide to better represent the polysaccharide.

Comments 11: Please ensure that all abbreviations are clearly defined at their first occurrence in the text and used consistently throughout the manuscript.

Response 11:We have corrected the acronyms throughout the manuscript and added a list of acronyms at the end of the article.

Reviewer 2 Report

Comments and Suggestions for Authors

This work focuses on reviewing the hypoglycemic mechanism of NP, focusing on the structure-activity relationship, gut flora-metabolism-immunity axis and key signaling pathways.

The manuscript is well written and the research topic is interesting. Particularly, they present an exhaustive discussion about the mechanisms involved in the hypoglycemic effect of NP polysaccharides, particularly emphasising the regulation of key signalling pathways and molecular targets. The informartion presented in this review provide a Good reference for further studies which could contribute to the developing of potential therapeutic agents base don natural polysaccharides.

There are some aspects that the authors should attend before the manuscript can be accepted for publication in Molecules.

  • Please define the following abbreviatures the first time they appear in the manuscript: FFA, VDCC, GS, LBP.
  • Subsections 4 and 5 have the same title.
  • Line 113: change “form” to “from”
  • I recommend to use “gut microbiota” instead of “intestinal flora”.
  • Please revise paragraph (lines 143-146), redaction is confusing.
  • Line 278: … is one of the most important ….
  • Line 354: Please change “Inhibits” to “Inhibition of”
  • Table 1. Please revise the organizarion and headings of table.

What do you mean with style of expression (grammar)?

Change “activation” instead of “activate”

Define DP

  • Line Please change “NANPH” to NADPH”

Author Response

Response to Reviewer 2 Comments

Dear Reviewers,

Thank you for your positive comments and constructive feedback on our manuscript. We highly value your insights and have carefully addressed all the points raised. The revisions we made are highlighted in red throughout the manuscript, and we have also added new sections and content where necessary. Below, we provide a detailed Response  to each of your comments.

Comments 1: Please define the following abbreviatures the first time they appear in the manuscript: FFA, VDCC, GS, LBP.

Response 1:We've corrected the manuscript's abbreviations and compiled a list at the end. Modified content is highlighted in red. For instance, "Free fatty acid (FFA)" at line 364, "Lycium barbarum polysaccharide (LBP)" at line 171, "Voltage-dependent calcium channel (VDCC)" at line 623, and "Glycogen synthase (GS)" at line 74.

Comments 2: Subsections 4 and 5 have the same title.

Response 2:We sincerely apologize for the oversight. We have renamed the duplicate subsection titles as follows: Subsection 4 is now "4. Regulation of Metabolism," and Subsection 5 is now "5. Regulation of Immunity."

Comments 3: Line 113: change “form” to “from”

Response 3:We have corrected the sentence "Selenium polysaccharide from sweet corn cob exhibits more potent glucose metabolism intervention than metformin" to "Sweet corn cob selenium polysaccharide decreased gut barrier permeability, modulated gut microbiota structure, and increased Firmicutes abundance" (lines 526–529).

Comments 4: I recommend to use “gut microbiota” instead of “intestinal flora”.

Response 4:We’ve revised the text to use “gut microbiota” instead of “intestinal flora” and “gut” instead of “intestinal.” For example, at lines 390 - 394, we now state: “Gut microbiota affects gut barrier function, body immunity and glycolipid metabolism. Disturbances in gut microbiota lead to a decrease in SCFAs...”

Comments 5: Please revise paragraph (lines 143-146), redaction is confusing.

Response 5:We have revised it to: "Gut microbiota affects gut barrier function, body immunity, and glycolipid metabolism. Disturbances in gut microbiota lead to a decrease in SCFAs, which triggers abnormal metabolism of BAs, branched-chain amino acids (BCAAs), and lipopolysaccharides (LPS), which in turn leads to T2D through IR and chronic inflammation" (lines 390–394).

Comments 6: Line 278: … is one of the most important ….

Response 6:We have corrected "Glucose Transporter (GLUT) is one of the important" to "Glucose Transporter (GLUT) is one of the most important" (line 618).

Comments 7: Line 354: Please change “Inhibits” to “Inhibition of”

Response 7:We have changed "4.3. Inhibits sugar digestion" to "4.3. Inhibition of sugar digestion".

Comments 8:Table 1. Please revise the organizarion and headings of table. What do you mean with style of expression (grammar)? Change “activation” instead of “activate”? Define DP  ?

Response 8:We have removed (grammar) and made changes to the title and structure of the table. The title of Table 2 has been changed to Regulatory Effects and Mechanisms of NP on Key Enzymes in Blood Glucose Metabolism, and we have changed “activation” instead of “activate” and simultaneously changed inhibitory to Inhibit. “activation” instead of ‘activate’ and synchronized the change from inhibitory to Inhibit. Due to an oversight on our part, the DP in the table should have been Natural Polysaccharides (NP), which we have corrected. We have optimized the text of the table to make it more understandable.

Comments 9: Line Please change “NANPH” to NADPH”

Response 9:We have changed “NANPH” to “NADPH” and highlighted the changes in red in the text.

Reviewer 3 Report

Comments and Suggestions for Authors

The submitted manuscript reviews the hypoglycemic mechanisms of natural polysaccharides, highlighting pathways such as PI3K/AKT and AMPK. However, it presents a redundant structure, has little translational discussion and dense writing.

1. Much of the information is redundant and fragmented, for example, inflammatory aspects are repeatedly touched upon in the manuscript and only attempted to be integrated in Figure 3.

2. I believe that the manuscript would be more interesting if the authors included a section on the clinical relevance and feasibility of its applications in the clinic.

3. In Figures 2 and 3, the authors suggest some probable mechanisms. However, it is not clear how the NPs are introduced into cells. Even though they themselves state that their intestinal absorption is poor.

4. Why did the authors name section 5 and 5 the same? Regulation of metabolism.

5. It is necessary that the authors select their references better, for example, number 1 is an evaluation of a polysaccharide, but the authors use it to mention the prevalence of diabetes, even though there is a large number of papers that deal with this subject.

6. The nomenclature of diabetes should be changed, according to the latest recommendations of the American Diabetes Association, it is no longer called Diabetes Mellitus type 2, now it is only Diabetes type 2.

7. Change the scientific names of plants and microorganisms mentioned in the manuscript to italics.

8. The nomenclature of the molecules should be taken care of, for example, NANPH is not correct. Correct.

Author Response

Response to Reviewer 3 Comments

Dear Reviewers,

Thank you for your positive comments and constructive feedback on our manuscript. We highly value your insights and have carefully addressed all the points raised. The revisions we made are highlighted in red throughout the manuscript, and we have also added new sections and content where necessary. Below, we provide a detailed Response  to each of your comments.

Comments 1: Much of the information is redundant and fragmented, for example, inflammatory aspects are repeatedly touched upon in the manuscript and only attempted to be integrated in Figure 3.

Response 1:We sincerely appreciate your insightful comments on the redundancy and dispersion of information in our manuscript, particularly regarding the integration of inflammation-related content.

In Response , we have completely restructured the discussion on inflammation, consolidating it primarily within the "Regulation of Immunity" chapter. We have introduced new content on the impact of hyperglycemic environments and excess fatty acids, and elaborated on the mechanisms through which NPs exert their effects. Redundant information has been removed, and we have cross-referenced relevant sections to clarify how chronic inflammation creates a vicious cycle through IRS-1 serine phosphorylation (leading to IR), mitochondrial ROS accumulation (worsening oxidative stress), and gut barrier damage (facilitating LPS entry into the bloodstream). We have emphasized how NPs can interrupt this cycle via multiple pathways. The mechanisms of inflammation previously scattered across sections on gut microbiota, metabolic regulation, and islet protection (such as TLR4/NF-κB, NLRP3 inflammasome, and JNK/IRS-1 axis) are now centralized to avoid repetition. These revisions are detailed in lines 964 - 1054.

Furthermore, we have dedicated a separate subsection "3.2. Gut Microbiota-Immunity Axis Regulatory Network" to discuss the interplay between gut microbiota and inflammation. We have also refined Figure 3 by eliminating redundancy and focusing on NP's regulatory effects, using "influence" to denote relationships for simplicity. Descriptive text has been added above the figure to enhance comprehension. The updated figure can be found at line 1056 on page 24.

Comments 2: I believe that the manuscript would be more interesting if the authors included a section on the clinical relevance and feasibility of its applications in the clinic.

Response 2:We have added a new chapter on "6. Clinical Application Strategies" which comprises three subsections: 6.1 Delivery System, 6.2 Design of Combination Drug Programs, and 6.3 Individualized Treatment Strategies. Each subsection offers specific clinical recommendations. These can be found at lines 1105 - 1109 on page 25, lines 1139 - 1143 on page 26, and lines 1153 - 1157 on page 26. To enhance clinical feasibility, we have also supplemented the text with new references.

Comments 3: In Figures 2 and 3, the authors suggest some probable mechanisms. However, it is not clear how the NPs are introduced into cells. Even though they themselves state that their intestinal absorption is poor.

Response 3:We have revised Figures 2 and 3 and elaborated on the pathways through which NPs exert hypoglycemic activity in the human body within both the Introduction and the molecular weight chapters. We also added more detailed introductory descriptions before Figures 2 and 3 to enhance their clarity. Additionally, the Conclusion section summarizes that low-molecular-weight NPs can directly target pathways such as α-glucosidase, PI3K/AKT, and AMPK to rapidly inhibit glucose absorption and promote glycogen synthesis. In contrast, high-molecular-weight NPs rely more on gut microbiota fermentation to generate SCFAs, enabling long-acting regulation of GLP-1 secretion and insulin sensitivity. We described the pathways through which NPs exert their activity in the Introduction (lines 78–87). For example: *"However, NPs can exert hypoglycemic functions by regulating gut microbiota, interfering with host metabolism, or directly targeting signaling pathways."*

Comments 4: Why did the authors name section 5 and 5 the same? Regulation of metabolism.

Response 4:We sincerely apologize for the oversight. We have renamed the duplicate subsection titles as follows: Subsection 4 is now "4. Regulation of Metabolism," and Subsection 5 is now "5. Regulation of Immunity."

Comments 5: It is necessary that the authors select their references better, for example, number 1 is an evaluation of a polysaccharide, but the authors use it to mention the prevalence of diabetes, even though there is a large number of papers that deal with this subject.

Response 5:We have replaced Reference 1 with the original reference directly citing this data (lines 1242–1244). The literature is as follows: Zhou, B.; Rayner, A.W.; Gregg, E.W.; Sheffer, K.E.; Carrillo-Larco, R.M.; Bennett, J.E.; Shaw, J.E.; Paciorek, C.J.; Singleton, R.K.; Pires, A.B.; *et al.* Worldwide trends in diabetes prevalence and treatment from 1990 to 2022: a pooled analysis of 1108 population-representative studies with 141 million participants. *Lancet* **2024**, *404*, 2077–2093. https://doi.org/10.1016/S0140-6736(24)02317-1.

Comments 6: The nomenclature of diabetes should be changed, according to the latest recommendations of the American Diabetes Association, it is no longer called Diabetes Mellitus type 2, now it is only Diabetes type 2.

Response 6:We corrected Diabetes type 2.(T2D) (line 63), and likewise Diabetes type 1 (T1D) (line 451).

Comments 7: Change the scientific names of plants and microorganisms mentioned in the manuscript to italics.

Response 7:We have italicized the scientific names of plants and microorganisms mentioned in the manuscript and highlighted them in red font.

Comments 8: The nomenclature of the molecules should be taken care of, for example, NANPH is not correct. Correct.

Response 8:We have changed “NANPH” to “NADPH” and highlighted the changes in red in the text.

Round 2

Reviewer 1 Report

Comments and Suggestions for Authors

After reviewing the revised manuscript, I find that the authors have addressed most of my comments and recommendations. The manuscript has been substantially improved, and I consider its current form suitable for publication.

Best wishes,

Reviewer 3 Report

Comments and Suggestions for Authors

The authors have addressed my observations adequately.